# Distributionally Robust Optimization Leads to Better Generalization: on SGD and Beyond

## Abstract

In this paper, we adopt distributionally robust optimization (DRO) (Ben-Tal et al., 2013) in hope to achieve a better generalization in deep learning tasks. We establish the generalization guarantees and analyze the localized Rademacher complexity for DRO, and conduct experiments to show that DRO obtains a better performance. We reveal the profound connection between SGD and DRO, i.e., selecting a batch can be viewed as choosing a distribution over the training set. From this perspective, we prove that SGD is prone to escape from bad stationary points and small batch SGD outperforms large batch SGD. We give an upper bound for the robust loss when SGD converges and keeps stable. We propose a novel Weighted SGD (WSGD) algorithm framework, which assigns high-variance weights to the data of the current batch. We devise a practical implement of WSGD that can directly optimize the robust loss. We test our algorithm on CIFAR-10 and CIFAR-100, and WSGD achieves significant improvements over the conventional SGD.

## 1 Introduction

In recent years, supervised learning based on deep neural networks (DNNs) has achieved state-of-the-art performances in various tasks of different domains, such as computer vision and natural language processing (Goodfellow et al., 2016). However, theoretical explanations for the generalization ability of deep learning remain challenging. Roughly speaking, supervised learning aims to learn a model from the training data. Let $\Theta$ denote the parameter space of the model, and $\mathcal{Z}$ denote the sample space. Let $\ell(z, \theta) : \mathcal{Z} \times \Theta \to \mathbb{R}_+$ denote the loss function related to the parameter $\theta$ and the instance $z$. In the general setting of supervised learning, there is a fixed but unknown distribution over $\mathcal{Z}$ (denoted $P$). We attempt to find a $\theta \in \Theta$, which gives a small generalization error. That is, we address the following optimization problem:

$$\min_{\theta \in \Theta} R(\theta) \triangleq \mathbb{E}_P \ell(z, \theta). \tag{1}$$

However, the real distribution $P$ is unknown, so we consider an empirical distribution. Given the training samples $S = (z_1, z_2, \cdots, z_m)$, the empirical distribution is defined as $\widehat{P}_m(z) = \frac{1}{m} \sum_{i=1}^m \mathbb{I}_{[z=z_i]}$, where $\mathbb{I}$ is the indicator function. The loss related to this distribution is called the empirical loss, which is

$$\widehat{R}(\theta) \triangleq \mathbb{E}_{\widehat{P}_m} \ell(z, \theta) = \frac{1}{m} \sum_{i=1}^m \ell(z_i, \theta), \tag{2}$$

and we minimize the empirical loss instead. This procedure is referred to as Empirical Risk Minimization (ERM).

The overwhelming capacity and the paucity of data make deep learning architectures susceptible to over-fitting. This urges us to propose a better approach to the minimization problem (1). Suppose there is a distribution family $\mathcal{M} = \{P_\lambda : \lambda \in \Lambda\}$ parametrized by $\lambda$, and the underlying distribution $P = P_{\lambda_0} \in \mathcal{M}$, where $\lambda_0$ is fixed but unknown. It is a common case in deep learning tasks that the structure of the data is complicated while the amount of data is small. Hence, when we use an estimator $\hat{\lambda}$ to estimate the parameter $\lambda_0$, the variance of $\hat{\lambda}$ would be quite large. Thus, rather than directly minimizing the true expected loss $R(\theta) = \mathbb{E}_{P_{\lambda_0}} \ell(z, \theta)$, we minimize a distributionally

robust loss alternatively, which can ensure the true expected loss to be small with high probability. The distributionally robust loss is defined as:

$$R(\theta, K) \triangleq \sup_{\lambda \in K} \mathbb{E}_{P_\lambda} \ell(z, \theta),$$

where $K \subset \Lambda$ is a neighbourhood of the estimator $\hat{\lambda}$. This procedure is called Distributionally Robust Optimization (DRO). Because we only have access to the given training dataset, we construct an unbiased[1] estimation of the expected loss $\mathbb{E}_{P_\lambda} \ell(z, \theta)$:

$$\frac{1}{m} \sum_{i=1}^{m} \frac{P_\lambda(z_i)}{P_{\lambda_0}(z_i)} \ell(z_i, \theta).$$

This can be viewed as the expected loss with respect to a perturbed empirical generalized distribution[2] $\hat{P}_\lambda = \frac{1}{m} \sum_{i=1}^{m} \frac{P_\lambda(z)}{P_{\lambda_0}(z)} \mathbb{I}_{[z=z_i]}$, so the DRO problem becomes minimizing the empirical distributionally robust loss:

$$\hat{R}(\theta, K) \triangleq \sup_{\lambda \in K} \mathbb{E}_{\hat{P}_\lambda} \ell(z, \theta) = \sup_{\lambda \in K} \sum_{i=1}^{m} \frac{P_\lambda(z_i)}{m P_{\lambda_0}(z_i)} \ell(z_i, \theta).$$

Keeping the motivations above in mind, we focus our attention on the **robust loss**, defined as the maximal possible weighted loss over a pre-specified weight set $\mathcal{P}$ for simplicity:

$$\hat{R}_{S,\mathcal{P}}(\theta) \triangleq \sup_{\boldsymbol{p} \in \mathcal{P}} \sum_{i=1}^{m} p_i \ell(z_i, \theta).$$

In deep learning, the high capacity of the model and the non-convexity of the objective function lead to an extremely complicated landscape and induce a lot of minima, and different minima have different generalization performances. In this case, an optimization algorithm is both an optimizer and a minimum selector, and the minimum preference of the optimization algorithm is crucial in deep learning. Stochastic Gradient Descent (SGD) and its variants are widely used to minimize the empirical loss (2). The simplest mini-batch SGD consists of iterative sampling of a batch of data and then updating the parameter:

$$\theta_{t+1} = \theta_t - \eta_t \Big( \frac{1}{|B_t|} \sum_{z_i \in B_t} \nabla_\theta \ell(z_i, \theta_t) \Big),$$

where $B_t \subset S$ is the sampled mini-batch, $\eta_t$ is the learning rate, and $|B_t|$ denotes the batch size. We interpret SGD from another perspective: at each iteration, SGD randomly chooses an empirical distribution over the data and performs a gradient descent update w.r.t. the corresponding loss. This interpretation shows a profound connection between SGD and DRO. Motivated by the connection, we develop a brand new version of SGD that can achieve a better generalization performance. The main contribution of this paper is summarized as follows:

- We clarify the connection between mini-batch SGD and DRO. We show that SGD helps escape from bad stationary points and the robust loss of local minima where SGD converges is not large.

- We analyze the generalization property of robust loss and show the trade off: the robust loss generalizes well when $\mathcal{P}$ is relatively small, but a larger $\mathcal{P}$ is beneficial to reducing the localized Rademacher complexity.

- We propose a framework of improved SGD algorithm, which enables the algorithm to explore more empirical distributions over the data. We give a practical improved SGD algorithm that can directly optimize the robust loss.

---

[1]To make the fraction reasonable, we assume $P_{\lambda_0}(z) > 0$ whenever $P_\lambda(z) > 0$.

[2]Strictly speaking, this is not a distribution because $\hat{P}_\lambda(\mathcal{Z})$ may not equals to 1, but its expectation is 1 (the expectation is taken over the sample $S$). Nonetheless, we can view $\hat{P}_\lambda$ as a measure over $\mathcal{Z}$.

## 2 RELATED WORK

SGD is widely used in large-scale machine learning problems. The convergence theory of SGD in convex cases is solid (Ghadimi & Lan, 2013; Shamir & Zhang, 2013). However, due to the non-convexity of deep learning models, the behaviour of SGD is more complicated. Dauphin et al. (2014) argued that it is the prevalence of saddle points rather than local minima that causes the difficulty in high dimensional non-convex optimization. Ge et al. (2015) analyzed the property of perturbed GD (i.e., gradient descent with an isotropic noise injected). They showed that perturbed GD has the ability to escape from saddle points in a polynomial number of iterations, and then converges to a second-order stationary point.

It is also well known that adding perturbation and noise at different levels of the training processes helps generalization, such as dropout (Srivastava et al., 2014), and random noise injection to the input data (Chapelle et al., 2001). The good performance of SGD might benefit from the noise. Keskar et al. (2016) discovered that large-batch training will incur a degradation in the generalization performance of the model. They explained this phenomenon with the concept of *sharp minima*, and carried out numerical experiments to support the idea that SGD with large batch size tends to converge to sharp minima, which leads to a bad generalization ability. However, Dinh et al. (2017) provided theoretical results indicating that most existing definitions of *sharpness* and *flatness* of minima can not be applied to some commonly used non-negative homogeneous activation functions such as ReLU.

DRO was originally applied to decision making under uncertainty. Namkoong & Duchi (2017) introduced DRO to machine learning, and viewed it as a variance-based regularization that allows us to automatically trade-off between bias and variance. They proved an $O(\frac{1}{m})$ approximation to the generalization error $R(\theta)$, in contrast to the general $O(\frac{1}{\sqrt{m}})$ approximation of the empirical risk. Recently, DRO is used to achieve fairness without demographic information in fair machine learning (Hashimoto et al., 2018).

In learning theory, Rademacher complexity is widely used to measure the complexity of a hypothesis set (Bartlett & Mendelson, 2002). Different from the general Rademacher complexity which considers the entire hypothesis set, localized Rademacher complexity only cares about the near optimal hypothesis (Bartlett et al., 2002). In fact, the hypothesis selected by a learning algorithm usually has better performance than the worst hypothesis. Thus, localized Rademacher complexity is usually much smaller than the general Rademacher complexity.

## 3 NOTATION AND PRELIMINARIES

In our work, we assume the loss function $\ell(z, \theta)$ is bounded between 0 and 1, and is continuously differentiable. Let $\Sigma$ denote the hyperplane $\{\boldsymbol{p} : \sum_{i=1}^{m} p_i = 1\}$ and $\Sigma_+$ denote the simplex $\{\boldsymbol{p} : \sum_{i=1}^{m} p_i = 1, \boldsymbol{p} \geq \boldsymbol{0}\}$. Denote $\boldsymbol{p}_0 = (\frac{1}{m}, \ldots, \frac{1}{m})^\top$. For $\boldsymbol{p} = (p_1, p_2, \ldots, p_m)^\top \in \Sigma$, let $\widehat{R}_{S,\boldsymbol{p}}(\theta)$ denote $\sum_{i=1}^{m} p_i \ell(z_i, \theta)$. Then the ERM loss can be written as $\widehat{R}_{S,\boldsymbol{p}_0}(\theta)$, and the robust loss w.r.t. the weight set $\mathcal{P}$ can be viewed as:

$$\widehat{R}_{S,\mathcal{P}}(\theta) = \sup_{\boldsymbol{p} \in \mathcal{P}} \widehat{R}_{S,\boldsymbol{p}}(\theta).$$

Let $\mathrm{conv}(\mathcal{P})$ denote the convex hull of $\mathcal{P}$. Given a $\boldsymbol{v} \in \Sigma$, let $\sigma(\boldsymbol{v})$ be the set of all the vectors whose entries are permutations of $\boldsymbol{v}$'s. Let $\mathcal{P}(\boldsymbol{v})$ denote the convex hull of $\sigma(\boldsymbol{v})$.

We define $\mathcal{P}(k)$ as the set of all the vectors that have $k$ $\frac{1}{k}$'s and $(m - k)$ 0's:

$$\mathcal{P}(k) = \left\{ \boldsymbol{p} = \frac{1}{k}\boldsymbol{v} : \boldsymbol{v} \in \{0, 1\}^m, \sum_{i=1}^{m} v_i = k \right\}.$$

Clearly, $\mathcal{P}(k)$ has a natural connection to SGD with batch size $k$: for any batch sampled from the dataset, there is a corresponding $\boldsymbol{p} \in \mathcal{P}(k)$, where $p_i = \frac{1}{k}$ if the $i$-th instance is in the batch.

In the remainder of the paper, we assume that $\mathcal{P}$ is *symmetric*, i.e., if $\boldsymbol{p} \in \mathcal{P}$, then $\sigma(\boldsymbol{p}) \subset \mathcal{P}$. This follows that $\boldsymbol{p}_0 \in \mathrm{conv}(\mathcal{P})$. And we assume that $\mathcal{P}$ is a compact set. We assume the entries of

$p \in \mathcal{P}$ sums to 1, but we relax the restrict that each entry of $p \in \mathcal{P}$ is non-negative, i.e., we assume $\mathcal{P} \subset \Sigma$.

To describe the properties of the set $\mathcal{P}$, we use the notation $\mathrm{RAD}_p(\mathcal{P})$ to denote the minimum radius of the $L_p$ ball centered at $p_0$ that contains $\mathcal{P}$, and we use $\mathrm{rad}_p(\mathcal{P})$ to denote the maximum radius of the $L_p$ ball centered at $p_0$ whose intersection with $\Sigma$ is contained within $\mathcal{P}$. We denote $\|\mathcal{P}\|_p = \sup_{p \in \mathcal{P}} \|p\|_p$, and $\|\mathcal{P}\|_0$ as the maximum number of non-zero entries of $p \in \mathcal{P}$.

**Definition 1.** *A function $f : \Theta \to \mathbb{R}$ is $L$-Lipschitz, if:*

$$|f(\theta_1) - f(\theta_2)| \le L\|\theta_1 - \theta_2\|, \forall \theta_1, \theta_2 \in \Theta.$$

**Definition 2.** *A function $f : \Theta \to \mathbb{R}$ is $\mu$-strongly convex if $f(\theta) - \frac{\mu}{2}\|\theta\|^2$ is convex.*

Some basic properties of the robust loss $\widehat{R}_{S,\mathcal{P}}(\theta)$ are summarized in Appendix A.

## 4 CONNECTION BETWEEN SGD AND DRO

SGD has long been regarded as an excellent optimizer. It is generally believed that the good performance of SGD is due to the noise introduced by stochastic gradient. But the existing results commonly view stochastic gradient as an oracle that provides an unbiased estimation of the true gradient, ignoring the fact that the noise is data-dependent. In this section, we directly analyze the good properties of mini-batch SGD from the distributionally robust perspective.

At each iteration, SGD calculates the gradient of $\widehat{R}_{S,p}(\theta)$, where $p$ is the weight vector corresponding to the current batch, and performs the update $\theta \leftarrow \theta - \eta \nabla \widehat{R}_{S,p}(\theta)$. This can be viewed as changing the distribution over the data. Thus SGD can improve the distributional robustness of the model. The following theorem shows that changing the distribution helps mini-batch SGD escape from some bad stationary points, which is characterized by the heterogeneity of gradients of each individual instance.

**Theorem 1.** *Let $S$ be a fixed sample of size $m$. For a fixed $\theta \in \Theta$, define a matrix $G = (\nabla_\theta \ell(z_1, \theta), \cdots, \nabla_\theta \ell(z_m, \theta))$, whose $i$-th column is the gradient of the $i$-th loss function w.r.t. $\theta$. For SGD with batch size $k$, denote $p$ as the random weight vector uniformly sampled from $\mathcal{P}(k)$. Then the expected squared update amount can be calculated by:*

$$\mathbb{E}_{p} \|\nabla \widehat{R}_{S,p}(\theta)\|_2^2 = \|\nabla \widehat{R}_{S,p_0}(\theta)\|_2^2 + \frac{m-k}{k(m-1)} \left( \frac{\mathrm{tr}(G^\top G)}{m} - \|\nabla \widehat{R}_{S,p_0}(\theta)\|_2^2 \right).$$

When $\frac{\mathrm{tr}(G^\top G)}{m} \gg \|\nabla \widehat{R}_{S,p_0}(\theta)\|_2^2$, the gradient of the ERM loss is relatively small, but the average squared length of the gradients $g_1, \cdots, g_m$ is large. This happens only if those gradients lie in some opposite directions and cancel with each other. In this case, if we leave out the instance with the largest gradient norm, then the average gradient of the rest of the sample would be almost as large as the $\frac{1}{m}$ of the largest gradient norm, and the local landscape of the average loss function would shift dramatically. This indicates the model pays too much attention to this specific instance and fails to learn the intrinsic structure of the data, which can be viewed as a signal of over-fitting. We call this a bad stationary point. So the scale of $\frac{\mathrm{tr}(G^\top G)}{m}$ and $\|\nabla \widehat{R}_{S,p_0}(\theta)\|_2^2$ captures the over-fitting tendency of a model by measuring the heterogeneity of gradients of the individual instance.

Furthermore, we can know from Theorem 1 that for any fixed $\theta$, $\mathbb{E}_{p}\|\nabla \widehat{R}_{S,p_0}(\theta)\|_2^2$ decreases monotonically when $k$ becomes larger, indicating the update amount of small batch SGD is larger than that of large batch SGD in expectation. Therefore, small batch SGD has a stronger ability to capture the non-uniformity of gradients and escape from bad stationary points.

Intuitively, if SGD converges and keeps stable when the distribution changes, the weighted average loss $\widehat{R}_{S,p}$ will not be too large, otherwise SGD has the tendency to escape. To theoretically prove it, we need to focus our attention to some simple case. If SGD has escaped from bad stationary points and the deep learning model really captures some intrinsic structure of the data, then it is conceivable that a batch of the instances $B_t$ is able to reflect some common features of the entire sample $S$, so the local landscape of the average loss function will not shift too much when we only consider a batch of the data. In this case, we have the following theorem.

**Theorem 2.** *Assume the loss function is $L$-Lipschitz. Suppose the SGD algorithm with constant learning rate $\eta$ converges to and stays in $\mathcal{B}$, where $\mathcal{B} = \{\theta : \|\theta - \theta_0\|_2 \leq B\}$ is a ball that contains a minimum of ERM loss $\widehat{R}_{S,\boldsymbol{p}_0}$. Suppose that $\forall \boldsymbol{p} \in \mathcal{P}(k)$, $\mathcal{B}$ contains a local minimum of $\widehat{R}_{S,\boldsymbol{p}}$ and $\widehat{R}_{S,\boldsymbol{p}}$ is $\mu$-strongly convex on $\mathcal{B}$. Further assume that the minimum value of the batch average loss $\widehat{R}_{S,\boldsymbol{p}}$ in $\mathcal{B}$ is not too large, i.e., $\min_{\theta \in \mathcal{B}} \widehat{R}_{S,\boldsymbol{p}}(\theta) \leq \min_{\theta \in \mathcal{B}} \widehat{R}_{S,\boldsymbol{p}_0}(\theta) + \delta$. Then the robust loss $\widehat{R}_{S,\mathcal{P}(k)}(\theta)$ can be bounded by:*

$$\widehat{R}_{S,\mathcal{P}(k)}(\theta) \leq \min_{\theta' \in \mathcal{B}} \widehat{R}_{S,\boldsymbol{p}_0}(\theta') + \frac{2B^2}{\mu\eta^2} + 2LB + \delta, \quad \forall \theta \in \mathcal{B}.$$

## 5 THEORETICAL ANALYSIS

### 5.1 GENERALIZATION GUARANTEE OF DRO

In this subsection, we introduce the concept of *Robust Rademacher Complexity* and use it to derive the generalization guarantee for robust loss, which provides justifications for DRO. We bound the expectation of robust loss by empirical robust loss plus a complexity term with high probability. Then we utilize the notion of *covering number* to show that the generalization of robust loss is not too hard compared to that of ERM loss.

**Definition 3** (Empirical Robust Rademacher Complexity). *Let $S = (z_1, \cdots, z_m)$ be a fixed sample of size $m$. The empirical robust Rademacher complexity of the parameter space $\Theta$ and the weight set $\mathcal{P}$ w.r.t. the sample $S$ is:*

$$\widehat{\mathscr{R}}_S(\Theta, \mathcal{P}) = \mathbb{E}_{\boldsymbol{\sigma}} \left[ \sup_{\theta \in \Theta} \sup_{\boldsymbol{p} \in \mathcal{P}} \sum_{i=1}^{m} \sigma_i p_i \ell(z_i, \theta) \right],$$

*where $\boldsymbol{\sigma} = (\sigma_1, \ldots, \sigma_m)^\top$ is a random vector with independent entries uniformly from $\{-1, 1\}$.*

**Theorem 3.** *For any $\delta > 0$, with probability at least $1 - \delta$, the following holds for all $\theta \in \Theta$:*

$$\mathbb{E}_{S'} \widehat{R}_{S',\mathcal{P}}(\theta) \leq \widehat{R}_{S,\mathcal{P}}(\theta) + 2\widehat{\mathscr{R}}_S(\Theta, \mathcal{P}) + 3m\|\mathcal{P}\|_\infty \sqrt{\frac{1}{2m} \log \frac{2}{\delta}}.$$

For a fixed sample $S$ of size $m$, denote $\Theta(S) = \left\{ (\ell(z_1, \theta), \cdots, \ell(z_m, \theta))^\top : \theta \in \Theta \right\}$ and $\Theta(S, \mathcal{P}) = \left\{ (mp_1\ell(z_1, \theta), \cdots, mp_m\ell(z_m, \theta))^\top : \theta \in \Theta, \boldsymbol{p} \in \mathcal{P} \right\}$. For a norm $\| \cdot \|$ in $\mathbb{R}^m$, denote the dual norm by $\| \cdot \|_*$. For $\epsilon > 0$, let $\mathcal{N}(A, \| \cdot \|, \epsilon)$ denote the covering number of the set $A \in \mathbb{R}^m$, which is the minimum cardinality of the $\epsilon$-net w.r.t. the norm $\| \cdot \|$. Denote by $d_*(m)$ an upper bound of $\|\boldsymbol{\sigma}\|_* = \|(\sigma_1, \cdots, \sigma_m)^\top\|_*$.

**Theorem 4.** *For any $\epsilon > 0$, the empirical Rademacher complexity and the empirical robust Rademacher complexity can be respectively bounded by:*

$$\widehat{\mathscr{R}}_S(\Theta) \leq \sqrt{\frac{2\log \mathcal{N}(\Theta(S), \| \cdot \|, \epsilon)}{m}} + \frac{\epsilon d_*(m)}{m}, \tag{3}$$

*and*

$$\widehat{\mathscr{R}}_S(\Theta, \mathcal{P}) \leq (1 + \mathrm{RAD}_2(\mathcal{P})) \sqrt{\frac{2\log \mathcal{N}(\Theta(S, \mathcal{P}), \| \cdot \|, \epsilon)}{m}} + \frac{\epsilon d_*(m)}{m}. \tag{4}$$

Notice that $\Theta(S, \mathcal{P})$ is generated from $\Theta(S)$ by having all the points perturbed a little bit, so it is conceivable that these two sets are close, implying that their covering numbers are close. Specifically, if we define

$$\widetilde{d} = \sup_{\boldsymbol{x} \in \Theta(S, \mathcal{P})} \inf_{\boldsymbol{y} \in \Theta(S)} \|\boldsymbol{x} - \boldsymbol{y}\|$$

as the maximum distance of a point in $\Theta(S, \mathcal{P})$ deviating from the set $\Theta(S)$, then it follows that an $\epsilon$-net of $\Theta(S)$ is also an $(\epsilon + \widetilde{d})$-net of $\Theta(S, \mathcal{P})$. To compare the right-hand sides of the covering

number guarantees (3) and (4), we change $\epsilon$ to $\epsilon + \widetilde{d}$ in (4), and we have:

$$(1 + \text{RAD}_2(\mathcal{P}))\sqrt{\frac{2\log\mathcal{N}(\Theta(S,\mathcal{P}), \|\cdot\|, \epsilon+\widetilde{d})}{m}} + \frac{(\epsilon+\widetilde{d})d_*(m)}{m}$$

$$\leq (1 + \text{RAD}_2(\mathcal{P}))\sqrt{\frac{2\log\mathcal{N}(\Theta(S), \|\cdot\|, \epsilon)}{m}} + \frac{\epsilon d_*(m)}{m} + \frac{\widetilde{d}d_*(m)}{m}.$$

When we use the $L_p$ norm, one can easily show that $\widetilde{d} \leq m\text{RAD}_p(\mathcal{P})$, and $\|\boldsymbol{\sigma}\|_*$ can also be bounded as $\|\boldsymbol{\sigma}\|_* \leq m^{\frac{1}{q}} = m^{\frac{p-1}{p}}$, so $d_*(m) = m^{\frac{p-1}{p}}$. Then the extra term $\frac{\widetilde{d}d_*(m)}{m}$ can be bounded by $m^{1-\frac{1}{p}}\text{RAD}_p(\mathcal{P})$.

Those bounds quantify how the size of $\mathcal{P}$ will influence the generalization ability, and give a sufficient condition on the size of $\mathcal{P}$ to ensure that the covering number bounds are close. Conceivably, when $\mathcal{P}$ is sufficiently small, the generalization of robust loss is not too difficult. It is worth noticing that the inequality $\mathcal{N}(\Theta(S,\mathcal{P}), \|\cdot\|, \epsilon+\widetilde{d}) \leq \mathcal{N}(\Theta(S), \|\cdot\|, \epsilon)$ is quite loose when the set $\Theta(S)$ is some sort of "solid," and this is often the case in deep neural networks, whose capacities are thought to be very high. In one word, it is completely feasible to consider DRO when using DNNs.

## 5.2 Localized Rademacher Complexity

Next, we analyze the localized Rademacher complexity based on robust loss. Localized Rademacher complexity measures the complexity of near optimal hypotheses, which is much smaller than the entire hypothesis space. Here we assume the hypothesis set is restricted to a $L_2$ norm ball.[3]

DRO can be viewed as a variance-based regularization technique (Namkoong & Duchi, 2017). A low robust loss means a low variance of the losses $(\ell(z_1, \theta), \cdots, \ell(z_m, \theta))$, since otherwise one can choose a $\boldsymbol{p} \in \mathcal{P}$ that puts more weight on the instances with a large loss, and obtains a high robust loss. A larger size of $\mathcal{P}$ means a more radical way to assign weights, indicating that DRO with large $\mathcal{P}$ renders a stricter regularization on the variance of the losses. Rademacher complexity measures the ability of a hypothesis set to fit high variance random noises, and naturally a low variance of the losses means a small Rademacher complexity. The following theorem reveals the connection between Rademacher complexity and DRO, and provides a model-free bound on the localized empirical Rademacher complexity.

**Theorem 5.** *Assume the loss function is L-Lipschitz w.r.t. $\theta$ for all $z$. Consider the following hypothesis set:*

$$\Theta_c = \left\{ \theta : \mathbb{E}_S\widehat{R}_{S,\mathcal{P}}(\theta) \leq c, \|\theta\|_2 < r \right\}.$$

*For any $\epsilon > 0$, let $\mathcal{N}(\Theta_c, \epsilon)$ denote the covering number of $\Theta_c$. Then for all $0 < \delta < 1$, with probability $\geq 1 - \delta$, the localized empirical Rademacher complexity can be bounded as:*

$$\widehat{\mathscr{R}}_S(\Theta_c) \leq \frac{1}{\sqrt{m}}\left(1 + \frac{4}{\sqrt{m}\text{rad}_\infty(\text{conv}(\mathcal{P}))}\right)\left(c + \epsilon\|\mathcal{P}\|_0\|\mathcal{P}\|_\infty L + \|\mathcal{P}\|_\infty\sqrt{\frac{m}{2}\log\frac{\mathcal{N}(\Theta_c, \epsilon)}{\delta}}\right).$$

Conceivably, when $m$ and $c$ are fixed, increasing the size of $\mathcal{P}$ reduces the set $\Theta_c$. This bound shows that increasing the size of $\mathcal{P}$ also reduces the complexity of $\Theta_c$ under certain conditions. For example, let $\mathcal{P}$ be the intersection of $\Sigma$ and a $L_\infty$ ball with radius $\rho$ centered at $\boldsymbol{p}_0$. When $\rho$ is relatively small compared to $\frac{1}{m}$, increasing $\rho$ does not change $\|\mathcal{P}\|_\infty$ too much, but $\frac{1}{\text{rad}_\infty(\text{conv}(\mathcal{P}))}$ is inversely proportional to $\rho$. Therefore, increasing $\rho$ can reduce the scale of $\frac{\|\mathcal{P}\|_\infty}{\text{rad}_\infty(\text{conv}(\mathcal{P}))}$ dramatically.

## 6 Weighted SGD

Motivated by the DRO and the previous theoretical analyses, we propose a new variant of SGD algorithms in Algorithm 1. We will see that this variant is simple and practical for applications.

---

[3] When we train models with a constant weight decay $\lambda_{wd}$, generally the $L_2$ norm of final parameters will not excess $\sqrt{f(\boldsymbol{0})/\lambda_{wd}}$, where $f$ is the loss function. This is because when the algorithm minimizing $L(\theta) = f(\theta) + \lambda_{wd}\|\theta\|_2^2$ outputs $\theta'$, we generally have $L(\theta') \leq L(\boldsymbol{0})$, hence, $\lambda_{wd}\|\theta'\|_2^2 \leq f(\theta') + \lambda_{wd}\|\theta'\|_2^2 \leq f(\boldsymbol{0})$.

We have emphasized the significance of distributional robustness before. The critical factors that influence distributional robustness are the size of $\mathcal{P}$ and the value of the corresponding robust loss $\widehat{R}_{S,\mathcal{P}}(\theta)$. In order to obtain a more distributionally robust model, a direct approach is to increase the size of $\mathcal{P}$. In the conventional SGD, the weight of the data in the current batch is $\frac{1}{k}$, and the other is 0. Due to the limited computational resources, we still only have access to one mini-batch at each iteration, but we can assign high-variance weights to the instances of the mini-batch. This implies that our algorithm enjoys the idea behind importance sampling. Therefore, our algorithm explores a wider region of empirical distributions, yielding a more distributionally robust solution.

---

**Algorithm 1** Weighted Stochastic Gradient Descent (WSGD)

---

**Require:** Initial parameter: $\theta_0$, Weight Generator: $G$, Learning Rate: $\eta_t$, Total Iteration: $T$, Training Data: $S = \{z_1, z_2, \ldots, z_m\}$.
1: **for** $t = 1$ to $T$ **do**
2:     Select a mini-batch: $B_t = \{z_{j_1}, z_{j_2}, \ldots, z_{j_{|B_t|}}\}$
3:     Generate weight: $(w_1, w_2, \ldots, w_{|B_t|}) \leftarrow G$
4:     Calculate normalization factor: $W = \sum_{i=1}^{|B_t|} w_i$
5:     Calculate stochastic gradient: $g_t = \frac{1}{W} \sum_{i=1}^{|Bt|} w_i \nabla_\theta \ell(z_{j_i}, \theta_{t-1})$
6:     Update parameter: $\theta_t = \theta_{t-1} - \eta_t g_t$
7: **end for**
8: **return** $\theta_T$

---

We turn to the choices of the weight generator. In particular, we recommend two approaches:

- $G_1(q, r)$: $w_i = r$ if $\ell(z_{j_i}, \theta_{t-1})$ is among the $q|B_t|$ smallest elements of the losses $(\ell(z_{j_1}, \theta_{t-1}), \cdots, \ell(z_{j_{B_t}}, \theta_{t-1}))$, otherwise $w_i = 1$;

- $G_2(q, r)$: Randomly select $q|B_t|$ of the indices $\{j_i\}$ and set their weights to $r$. Set the others to 1,

where $q \in [0, 1]$ is the proportion of the instances whose weight is set to $r$, and $r$ is a pre-specified constant. Notice that $r$ is not necessarily non-negative.

We have conducted experiments to show that Algorithm 1 with generator $G_1$ achieves a better performance on classification tasks than the conventional SGD (see Section 7). We refer to Algorithm 1 with weight generator $G_1(q, r)$ as **WSGD(q, r)**.

Different from the conventional SGD, WSGD$(q, r)$ directly optimizes a robust loss. It is easy to prove that there exists a vector $\boldsymbol{v}(q, r) \in \Sigma$ such that the stochastic gradient in WSGD$(q, r)$ is an unbiased estimation of the gradient of $\widehat{R}_{S,\mathcal{P}(\boldsymbol{v}(q,r))}(\theta)$ at differentiable points (see Appendix C for details). Furthermore, the stochastic gradient in WSGD$(q, r)$ is bounded by the *Lipshitz* constant of $\ell$ when $r$ is non-negative. Taking advantages of previous work, we can give the convergence guarantee for WSGD$(q, r)$ with a non-negative $r$ in the strongly convex case:

**Corollary 1** (Shamir & Zhang 2013). *Suppose the loss function $\ell(z, \theta)$ is $\mu$-strongly convex and $L$-Lipshitz. Consider WSGD$(q, r)$ with step size $\eta_t = \frac{1}{\mu t}$ and $r$ is non-negative. Let $\widehat{R}^*_{S,\mathcal{P}(\boldsymbol{v}(q,r))}$ be the optimal value of $\widehat{R}_{S,\mathcal{P}(\boldsymbol{v}(q,r))}(\theta)$. Then for all $T > 1$, it holds that:*

$$\mathbb{E}[\widehat{R}_{S,\mathcal{P}(\boldsymbol{v}(q,r))}(\theta_T) - \widehat{R}^*_{S,\mathcal{P}(\boldsymbol{v}(q,r))}] \leq \frac{17L^2(1 + \log(T))}{\mu T}.$$

It is worth pointing out that $G_2$ usually yields a lower test loss in our trial. Strictly speaking, this means $G_2$ has a stronger generalization ability. However, the loss we choose in classification tasks is different from the true loss (0-1 loss), thus the algorithm with a low test loss not always has a low prediction error. We still recommend $G_2$ because it may achieve a good performance on tasks which we can deal with the true loss directly.

## 7 EXPERIMENTS

In this section, we compare WSGD with the conventional stochastic gradient descent, which is regarded as the optimizer with the best generalization. We carry out the experiment on the CIFAR-10 and CIFAR-100 datasets (Krizhevsky & Hinton, 2009), which both have 50k training data and 10k test data. On CIFAR-10, we apply our algorithm and SGD to two small networks and two large networks: ResNet-44, ResNet-56, VGG-16 and ResNet-34 (He et al., 2016; Simonyan & Zisserman, 2014). On CIFAR-100, we apply our algorithm and SGD to three large networks: VGG-16, ResNet-34 and DenseNet-121 (Huang et al., 2017). We train the networks following the data augmentation in Lee et al. (2015): 4 pixels are padded on each side, and a $32 \times 32$ crop is randomly sampled from the padded image or its horizontal flip. We only do evaluation on the original $32 \times 32$ image while testing. In order to compare the generalization ability of optimization algorithms, we train the models for 600 epochs to ensure convergence. We set the initial learning rate to $0.1$, and divide it by 10 at 300 and 450 epochs. The choice of hyper-parameters in $\text{WSGD}(q, r)$ is listed in the experiment results. Empirically speaking, the hyper-parameters are mainly related to the dataset and network architecture, and a smaller $r$ is suitable for larger networks.

We summarize the experiment results on CIFAR-10 and CIFAR-100 in Table 1. For each network and algorithm, we report the test accuracy, the $L_2$ ball and $L_\infty$ ball robust loss. $L_p$ ball robust loss is the robust loss with $\mathcal{P}$ as the intersection of $\Sigma_+$ and the $L_p$ ball centered at $\boldsymbol{p}_0$ with radius $\text{rad}_p(\Sigma_+)$. In other words, we consider the largest $L_p$ ball in $\Sigma$ centered $\boldsymbol{p}_0$ that has positive entries.

From table 1 we can see that WSGD outperforms SGD in all experiments. On CIFAR-10, WSGD decreases the misclassification rate by $\approx 5‰$ except VGG-16. On CIFAR-100, WSGD decreases the misclassification rate by $5‰$ to $15‰$. Moreover, the models trained by WSGD have a lower $L_2$ ball and $L_\infty$ ball robust loss than SGD. The good performance shows WSGD improves the distributional robustness of the models.

Table 1: Top 1 accuracy(%) and robust loss($\times 10^{-3}$) on CIFAR 10 and CIFAR 100

| Network | Params | Acc | $L_2$-ball | $L_\infty$-ball | Method | Dataset |
|---------|--------|-----|-----------|-----------------|--------|---------|
| ResNet-44 | 0.66M | 93.60 | 0.5902 | 1.116 | SGD | CIFAR-10 |
| ResNet-44 | 0.66M | **94.24** | 0.4892 | 0.9507 | WSGD(0.5,0.3) | CIFAR-10 |
| ResNet-56 | 0.86M | 93.86 | 0.3762 | 0.6745 | SGD | CIFAR-10 |
| ResNet-56 | 0.86M | **94.33** | 0.3070 | 0.5835 | WSGD(0.5,0.4) | CIFAR-10 |
| VGG-16 | 14.74M | 94.05 | 0.9617 | 1.112 | SGD | CIFAR-10 |
| VGG-16 | 14.74M | **94.30** | 0.8807 | 1.004 | WSGD(0.5,0.2) | CIFAR-10 |
| ResNet-34 | 21.30M | 95.13 | 0.7113 | 0.9379 | SGD | CIFAR-10 |
| ResNet-34 | 21.30M | **95.65** | 0.6422 | 0.8381 | WSGD(0.5,0) | CIFAR-10 |
| VGG-16 | 14.78M | 72.13 | 1.535 | 2.942 | SGD | CIFAR-100 |
| VGG-16 | 14.78M | **73.60** | 1.209 | 2.346 | WSGD(0.5,0) | CIFAR-100 |
| ResNet-34 | 21.35M | 77.89 | 1.473 | 2.562 | SGD | CIFAR-100 |
| ResNet-34 | 21.35M | **78.43** | 0.9876 | 1.871 | WSGD(0.5,0) | CIFAR-100 |
| DenseNet-121 | 7.13M | 79.31 | 3.359 | 4.614 | SGD | CIAFR-100 |
| DenseNet-121 | 7.13M | **80.35** | 1.196 | 2.152 | WSGD(0.5,-0.1) | CIAFR-100 |

## 8 DISCUSSION AND CONCLUSION

In this paper we have theoretically analyzed the good property of DRO, and revealed the profound connection between SGD and DRO. Accordingly we have proposed a practical algorithm that can utilize the non-isotropic noise of the stochastic gradient. We have tested WSGD algorithm on CIFAR-10 and CIFAR-100, achieving significant improvements compared to SGD.

We hope this paper can inspire works that theoretically study the generalization ability of optimization algorithms. We think one should pay attention to the loss of each individual instance, rather than the average of the entire sample. We expect future works on SGD without simplification to isotropic noise, and algorithms that take advantage of the non-isotropic noise of the stochastic gradient.

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

## A  SOME PROPERTIES OF ROBUST LOSS

**Lemma 1.** *Let $\ell_1, \cdots, \ell_m$ be arbitrary real numbers. Let $\mathrm{conv}(\mathcal{P})$ denote the convex hull of $\mathcal{P}$, then*

$$\sup_{\boldsymbol{p} \in \mathcal{P}} \sum_{i=1}^{m} p_i \ell_i = \sup_{\boldsymbol{p} \in \mathrm{conv}(\mathcal{P})} \sum_{i=1}^{m} p_i \ell_i.$$

**Lemma 2.** *Let $\ell_1, \cdots, \ell_m$ be arbitrary positive numbers. Assume $\mathcal{P} \in \Sigma$ and is symmetric. Then the following holds for all $\boldsymbol{p} \subset \mathcal{P}$:*

$$|\sum_{i=1}^{m} p_i \ell_i| \le \sup_{\boldsymbol{p}' \in \mathcal{P}} \sum_{i=1}^{m} p'_i \ell_i.$$

**Lemma 3.** *Assume the loss function $\ell(z, \theta)$ is L-Lipschitz with respect to $\theta$ for all $z$, then for any fixed sample $S$ of size $m$, $\widehat{R}_{S,\mathcal{P}}(\theta)$ is $\|\mathcal{P}\|_0 \|\mathcal{P}\|_\infty L$-Lipschitz*

*Proof.*

$$|\widehat{R}_{S,\mathcal{P}}(\theta) - \widehat{R}_{S,\mathcal{P}}(\theta')|$$
$$= |\sup_{\boldsymbol{p} \in \mathcal{P}} \sum p_i \ell(z_i, \theta) - \sup_{\boldsymbol{p} \in \mathcal{P}} \sum p_i \ell(z_i, \theta')|$$
$$\le \max \left\{ \sup_{\boldsymbol{p} \in \mathcal{P}} \sum p_i \left( \ell(z_i, \theta) - \ell(z_i, \theta') \right), \sup_{\boldsymbol{p} \in \mathcal{P}} \sum p_i \left( \ell(z_i, \theta') - \ell(z_i, \theta) \right) \right\}.$$

Notice that $|\ell(z_i, \theta) - \ell(z_i, \theta')| \le L\|\theta - \theta'\|$ for all $z_i$, we have:

$$|\sum p_i \left( \ell(z_i, \theta) - \ell(z_i, \theta') \right)|$$
$$\le \sum |p_i| L\|\theta - \theta'\|$$
$$\le \|\mathcal{P}\|_0 \|\mathcal{P}\|_\infty L\|\theta - \theta'\|.$$

One can develop various bounds on the *Lipschitz* constant of $\widehat{R}_{S,\mathcal{P}}(\theta)$. $\qquad\square$

**Lemma 4** (Convexity). *Assume that all entries of $\boldsymbol{p} \in \mathcal{P}$ is non-negative. Futher assume that $\ell(z, \theta)$ is convex with respect to $\theta$ for all $z$, then for a fix sample of size $m$, the robust loss $\widehat{R}_{S,\mathcal{P}}(\theta)$ is convex. And denote $\boldsymbol{p}^* = \arg\sup_{\boldsymbol{p} \in \mathcal{P}} \widehat{R}_{S,\boldsymbol{p}}(\theta)$, then for any $\boldsymbol{g}_i \in \partial_\theta \ell(z_i, \theta)$, $\sum_{i=1}^{m} p_i^* \boldsymbol{g}_i$ is in the subgradient of $\widehat{R}_{S,\mathcal{P}}(\theta)$ at point $\theta$.*

*Proof.* For any $\boldsymbol{p} \in \mathcal{P}$, $\widehat{R}_{S,\boldsymbol{p}}(\theta)$ is a convex combination of convex functions, thus it is convex. $\widehat{R}_{S,\mathcal{P}}$ is supremum over a family of convex functions, thus it is also convex. To complete the proof, by the definition of subgradient, we have the following holds for all $i = 1, \cdots, m$:

$$\ell(z_i, \theta') \ge \ell(z_i, \theta) + \langle \boldsymbol{g}_i, \theta' - \theta \rangle.$$

Then,

$$\widehat{R}_{S,\boldsymbol{p}^*}(\theta') = \sum_{i=1}^{m} p_i^* \ell(z_i, \theta')$$
$$\ge \sum_{i=1}^{m} p_i^* \ell(z_i, \theta) + \langle \sum_{i=1}^{m} p_i^* \boldsymbol{g}_i, \theta' - \theta \rangle.$$

So,

$$\widehat{R}_{S,\mathcal{P}}(\theta') \ge \widehat{R}_{S,\boldsymbol{p}^*}(\theta')$$
$$\ge \sum_{i=1}^{m} p_i^* \ell(z_i, \theta) + \langle \sum_{i=1}^{m} p_i^* \boldsymbol{g}_i, \theta' - \theta \rangle$$
$$= \widehat{R}_{S,\mathcal{P}}(\theta) + \langle \sum_{i=1}^{m} p_i^* \boldsymbol{g}_i, \theta' - \theta \rangle.$$

$\square$

**Lemma 5.** *Assume the loss function $\ell(z, \theta)$ is $\mu$-strongly convex (convex), then for any $\mathcal{P} \subset \Sigma_+$, $\widehat{R}_{S,\mathcal{P}}$ is $\mu$-strongly convex (convex).*

**Lemma 6.** *Assume the loss function $\ell(z, \theta)$ takes values in $[0, 1]$, then the robust loss $\widehat{R}_{S,\mathcal{P}}$ is bounded by $\frac{1 + \|\mathcal{P}\|_1}{2}$*

## B  PROOF OF THE THEOREMS

**Theorem 1.** *Let $S$ be a fixed sample of size $m$. For a fixed $\theta \in \Theta$, define a matrix $\boldsymbol{G} = (\nabla_\theta \ell(z_1, \theta), \cdots, \nabla_\theta \ell(z_m, \theta))$, whose $i$-th column is the gradient of the $i$-th loss function w.r.t. $\theta$. For SGD with batch size $k$, denote $\boldsymbol{p}$ as the random weight vector uniformly sampled from $\mathcal{P}(k)$. Then the expected squared update amount can be calculated by:*

$$\mathbb{E}_{\boldsymbol{p}} \|\nabla \widehat{R}_{S,\boldsymbol{p}}(\theta)\|_2^2 = \|\nabla \widehat{R}_{S,\boldsymbol{p}_0}(\theta)\|_2^2 + \frac{m-k}{k(m-1)} \left( \frac{\mathrm{tr}(\boldsymbol{G}^\top \boldsymbol{G})}{m} - \|\nabla \widehat{R}_{S,\boldsymbol{p}_0}(\theta)\|_2^2 \right).$$

*Proof.*

$$
\begin{aligned}
\mathbb{E}_{\boldsymbol{p}} \|\nabla \widehat{R}_{S,\boldsymbol{p}}(\theta)\|_2^2 &= \mathbb{E}_{\boldsymbol{p}} \|\boldsymbol{G}\boldsymbol{p}\|_2^2 \\
&= \mathbb{E}_{\boldsymbol{p}} \|\boldsymbol{G}(\boldsymbol{p} - \boldsymbol{p}_0)\|_2^2 + \|\boldsymbol{G}\boldsymbol{p}_0\|_2^2 \\
&= \mathbb{E}_{\boldsymbol{p}} (\boldsymbol{p} - \boldsymbol{p}_0)^\top \boldsymbol{G}^\top \boldsymbol{G}(\boldsymbol{p} - \boldsymbol{p}_0) + \|\boldsymbol{G}\boldsymbol{p}_0\|_2^2 \\
&= \mathbb{E}_{\boldsymbol{p}} \mathrm{tr}\left[ (\boldsymbol{p} - \boldsymbol{p}_0)^\top \boldsymbol{G}^\top \boldsymbol{G}(\boldsymbol{p} - \boldsymbol{p}_0) \right] + \|\boldsymbol{G}\boldsymbol{p}_0\|_2^2 \\
&= \mathbb{E}_{\boldsymbol{p}} \mathrm{tr}\left[ (\boldsymbol{p} - \boldsymbol{p}_0)(\boldsymbol{p} - \boldsymbol{p}_0)^\top \boldsymbol{G}^\top \boldsymbol{G} \right] + \|\boldsymbol{G}\boldsymbol{p}_0\|_2^2 \\
&= \mathrm{tr}\left[ \mathbb{E}_{\boldsymbol{p}} (\boldsymbol{p} - \boldsymbol{p}_0)(\boldsymbol{p} - \boldsymbol{p}_0)^\top \boldsymbol{G}^\top \boldsymbol{G} \right] + \|\boldsymbol{G}\boldsymbol{p}_0\|_2^2
\end{aligned}
$$

$$
\begin{aligned}
\mathbb{E}_{\boldsymbol{p}} (\boldsymbol{p} - \boldsymbol{p}_0)(\boldsymbol{p} - \boldsymbol{p}_0)^\top &= \mathbb{E}_{\boldsymbol{p}} \left[ \boldsymbol{p}\boldsymbol{p}^\top + \boldsymbol{p}_0\boldsymbol{p}_0^\top - \boldsymbol{p}\boldsymbol{p}_0^\top - \boldsymbol{p}_0\boldsymbol{p}^\top \right] \\
&= \mathbb{E}_{\boldsymbol{p}} \left[ \boldsymbol{p}\boldsymbol{p}^\top \right] - \boldsymbol{p}_0\boldsymbol{p}_0^\top
\end{aligned}
$$

One can easily calculate that

$$
\mathbb{E}_{\boldsymbol{p}} \left[ \boldsymbol{p}\boldsymbol{p}^\top \right] = \begin{pmatrix}
\frac{1}{mk} & \frac{k-1}{m(m-1)k} & \cdots & \frac{k-1}{m(m-1)k} \\
\frac{k-1}{m(m-1)k} & \frac{1}{mk} & \cdots & \frac{k-1}{m(m-1)k} \\
\vdots & & \ddots & \vdots \\
\frac{k-1}{m(m-1)k} & \cdots & \frac{k-1}{m(m-1)k} & \frac{1}{mk}
\end{pmatrix},
$$

when $\boldsymbol{p}$ is sampled from $\mathcal{P}(k)$, namely $k$ entries of $\boldsymbol{p}$ are randomly set to $\frac{1}{k}$, while other entries are 0. And

$$
\boldsymbol{p}_0\boldsymbol{p}_0^\top = \begin{pmatrix}
\frac{1}{m^2} & \frac{1}{m^2} & \cdots & \frac{1}{m^2} \\
\frac{1}{m^2} & \frac{1}{m^2} & \cdots & \frac{1}{m^2} \\
\vdots & & \ddots & \vdots \\
\frac{1}{m^2} & \cdots & \frac{1}{m^2} & \frac{1}{m^2}
\end{pmatrix},
$$

Then

$$\text{tr}\left[\mathbb{E}_{\boldsymbol{p}}(\boldsymbol{p}-\boldsymbol{p}_0)(\boldsymbol{p}-\boldsymbol{p}_0)^\top \boldsymbol{G}^\top \boldsymbol{G}\right]$$

$$= \left(\frac{1}{mk}-\frac{1}{m^2}\right)\text{tr}(\boldsymbol{G}^\top \boldsymbol{G}) + \left(\frac{k-1}{m(m-1)k}-\frac{1}{m^2}\right)\left(\mathbf{1}^\top \boldsymbol{G}^\top \boldsymbol{G}\mathbf{1} - \text{tr}(\boldsymbol{G}^\top \boldsymbol{G})\right)$$

$$= \left(\frac{1}{mk}-\frac{1}{m^2}\right)\left(\text{tr}(\boldsymbol{G}^\top \boldsymbol{G}) - \frac{\mathbf{1}^\top \boldsymbol{G}^\top \boldsymbol{G}\mathbf{1}}{m}\right) +$$

$$\left(\frac{k-1}{m(m-1)k}-\frac{1}{m^2}\right)\left(\frac{\mathbf{1}^\top \boldsymbol{G}^\top \boldsymbol{G}\mathbf{1}}{m} - \text{tr}(\boldsymbol{G}^\top \boldsymbol{G})\right)$$

$$= \frac{m-k}{k(m-1)}\left(\frac{\text{tr}(\boldsymbol{G}^\top \boldsymbol{G})}{m} - \boldsymbol{p}_0^\top \boldsymbol{G}^\top \boldsymbol{G}\boldsymbol{p}_0\right)$$

$\square$

**Theorem 2.** *Assume the loss function is L-Lipschitz. Suppose the SGD algorithm with constant learning rate $\eta$ converges to and stays in $\mathcal{B}$, where $\mathcal{B} = \{\theta : \|\theta - \theta_0\|_2 \leq B\}$ is a ball that contains a minimum of ERM loss $\widehat{R}_{S,\boldsymbol{p}_0}$. Suppose that $\forall \boldsymbol{p} \in \mathcal{P}(k)$, $\mathcal{B}$ contains a local minimum of $\widehat{R}_{S,\boldsymbol{p}}$ and $\widehat{R}_{S,\boldsymbol{p}}$ is $\mu$-strongly convex on $\mathcal{B}$. Further assume that the minimum value of the batch average loss $\widehat{R}_{S,\boldsymbol{p}}$ in $\mathcal{B}$ is not too large, i.e., $\min_{\theta \in \mathcal{B}} \widehat{R}_{S,\boldsymbol{p}}(\theta) \leq \min_{\theta \in \mathcal{B}} \widehat{R}_{S,\boldsymbol{p}_0}(\theta) + \delta$. Then the robust loss $\widehat{R}_{S,\mathcal{P}(k)}(\theta)$ can be bounded by:*

$$\widehat{R}_{S,\mathcal{P}(k)}(\theta) \leq \min_{\theta' \in \mathcal{B}} \widehat{R}_{S,\boldsymbol{p}_0}(\theta') + \frac{2B^2}{\mu\eta^2} + 2LB + \delta, \ \ \forall \theta \in \mathcal{B}.$$

*Proof.* We count the iteration after SGD converges to $\mathcal{B}$ and stays in $\mathcal{B}$. We denote $\theta_t$ the parameter at iteration $t$ and $\boldsymbol{p}^{(t)}$ the weight at iteration $t$.

Due to the $\mu$-strongly convexity of $\widehat{R}_{S,\boldsymbol{p}}(\theta), \forall \boldsymbol{p} \in \mathcal{P}(k)$ we have:

$$\frac{1}{2\mu}\|\nabla \widehat{R}_{S,\boldsymbol{p}^{(t)}}(\theta_t)\|_2^2 \geq \widehat{R}_{S,\boldsymbol{p}^{(t)}}(\theta_t) - \min_{\theta' \in \mathcal{B}} \widehat{R}_{S,\boldsymbol{p}^{(t)}}(\theta')$$

$$\geq \widehat{R}_{S,\boldsymbol{p}^{(t)}}(\theta_t) - \min_{\theta' \in \mathcal{B}} \widehat{R}_{S,\boldsymbol{p}_0}(\theta') - \delta.$$

Since the algorithm stays in $\mathcal{B}$, the norm of the stochastic gradient can be bounded by $\frac{2B}{\eta}$. Thus:

$$\widehat{R}_{S,\boldsymbol{p}^{(t)}}(\theta_t) \leq \min_{\theta' \in \mathcal{B}} \widehat{R}_{S,\boldsymbol{p}_0}(\theta') + \frac{2B^2}{\mu\eta^2} + \delta.$$

Plus, the weighted average loss $\widehat{R}_{S,\boldsymbol{p}^{(t)}}$ is *L-Lipschitz*, so $\widehat{R}_{S,\boldsymbol{p}^{(t)}}(\theta)$ can be bounded by:

$$\widehat{R}_{S,\boldsymbol{p}^{(t)}}(\theta) \leq \min_{\theta' \in \mathcal{B}} \widehat{R}_{S,\boldsymbol{p}_0}(\theta') + \frac{2B^2}{\mu\eta^2} + 2LB + \delta, \ \forall \theta \in \mathcal{B}.$$

As SGD algorithm keeps stable in $\mathcal{B}$, the above equality holds for all $\boldsymbol{p}^{(t)} \in \mathcal{P}(k)$. Finally we have:

$$\widehat{R}_{S,\mathcal{P}(k)}(\theta) \leq \min_{\theta' \in \mathcal{B}} \widehat{R}_{S,\boldsymbol{p}_0}(\theta') + \frac{2B^2}{\mu\eta^2} + 2LB + \delta, \ \forall \theta \in \mathcal{B}.$$

$\square$

**Definition 1** (Robust Rademacher Complexity). *For any $m \geq 1$, the robust Rademacher complexity of the parameter space $\Theta$ and the weight set $\mathcal{P}$ is defined as:*

$$\mathscr{R}_m(\Theta, \mathcal{P}) = \mathbb{E}_{S}\left[\widehat{\mathscr{R}}_S(\Theta, \mathcal{P})\right].$$

**Theorem 3.** *For any $\delta > 0$, with probability at least $1 - \delta$, the following holds for all $\theta \in \Theta$:*

$$\mathbb{E}_{S'} \widehat{R}_{S',\mathcal{P}}(\theta) \leq \widehat{R}_{S,\mathcal{P}}(\theta) + 2\widehat{\mathscr{R}}_S(\Theta, \mathcal{P}) + 3m\|\mathcal{P}\|_\infty \sqrt{\frac{1}{2m}\log\frac{2}{\delta}}.$$

*Proof.* For simplicity, let $R_{m,\mathcal{P}}(\theta) = \mathbb{E}_{S'}\left[\widehat{R}_{S',\mathcal{P}}(\theta)\right]$, where $S'$ is $m$ examples i.i.d. drawn from the underlying distribution. For a sample $S = (z_1, \cdots, z_m)$ of size $m$, denote

$$\Phi(S) = \sup_{\theta \in \Theta}\left[R_{m,\mathcal{P}}(\theta) - \widehat{R}_{S,\mathcal{P}}(\theta)\right].$$

If we change $z_i$ to $\widetilde{z}_i$ and get $\widetilde{S} = (z_1, \cdots, \widetilde{z}_i, \cdots, z_m)$, then

$$\begin{aligned}
\Phi(S) - \Phi(\widetilde{S}) &\le \sup_{\theta \in \Theta}\left[\widehat{R}_{\widetilde{S},\mathcal{P}}(\theta) - \widehat{R}_{S,\mathcal{P}}(\theta)\right] \\
&\le \sup_{\theta \in \Theta}\sup_{\boldsymbol{p} \in \mathcal{P}} p_i(\ell(z_i, \theta) - \ell(\widetilde{z}_i, \theta)) \\
&\le \|\mathcal{P}\|_\infty.
\end{aligned}$$

Similarly we have $|\Phi(S) - \Phi(\widetilde{S})| \le \|\mathcal{P}\|_\infty$. Then by McDiarmid's inequality(McDiarmid, 1989), for any $\delta > 0$, the following holds with probability at least $1 - \frac{\delta}{2}$:

$$\Phi(S) \le \mathbb{E}_S[\Phi(S)] + m\|\mathcal{P}\|_\infty\sqrt{\frac{1}{2m}\log\frac{2}{\delta}}.$$

Next we are to bound $\mathbb{E}_S[\Phi(S)]$. By symmetrization, we have:

$$\begin{aligned}
\mathbb{E}_S[\Phi(S)] &= \mathbb{E}_S\sup_{\theta \in \Theta}\left[R_{m,\mathcal{P}}(\theta) - \widehat{R}_{S,\mathcal{P}}(\theta)\right] \\
&= \mathbb{E}_S\sup_{\theta \in \Theta}\left[\mathbb{E}_{S'}\widehat{R}_{S',\mathcal{P}}(\theta) - \widehat{R}_{S,\mathcal{P}}(\theta)\right] \\
&\le \mathbb{E}_{S,S'}\sup_{\theta \in \Theta}\left[\widehat{R}_{S',\mathcal{P}}(\theta) - \widehat{R}_{S,\mathcal{P}}(\theta)\right] \\
&\le \mathbb{E}_{S,S'}\sup_{\theta \in \Theta}\sup_{\boldsymbol{p} \in \mathcal{P}}\left[\sum_{i=0}^{m} p_i(\ell(z_i, \theta) - \ell(z'_i, \theta))\right] \\
&= \mathbb{E}_{\boldsymbol{\sigma},S,S'}\sup_{\theta \in \Theta}\sup_{\boldsymbol{p} \in \mathcal{P}}\left[\sum_{i=0}^{m} \sigma_i p_i(\ell(z_i, \theta) - \ell(z'_i, \theta))\right] \\
&\le \mathbb{E}_{\boldsymbol{\sigma},S,S'}\sup_{\theta \in \Theta}\sup_{\boldsymbol{p} \in \mathcal{P}}\left[\sum_{i=0}^{m} \sigma_i p_i \ell(z_i, \theta)\right] + \mathbb{E}_{\boldsymbol{\sigma},S,S'}\sup_{\theta \in \Theta}\sup_{\boldsymbol{p} \in \mathcal{P}}\left[\sum_{i=0}^{m} -\sigma_i p_i \ell(z'_i, \theta)\right] \\
&= 2\mathscr{R}_m(\Theta, \mathcal{P}).
\end{aligned}$$

This shows that

$$\sup_{\theta \in \Theta}\left[R_{m,\mathcal{P}} - \widehat{R}_{S,\mathcal{P}}(\theta)\right] \le 2\mathscr{R}_m(\Theta, \mathcal{P}) + m\|\mathcal{P}\|_\infty\sqrt{\frac{1}{2m}\log\frac{2}{\delta}}$$

holds with probability at least $1 - \frac{\delta}{2}$.

To complete the proof, next we show that

$$\mathscr{R}_m(\Theta, \mathcal{P}) \le \widehat{\mathscr{R}}_S(\Theta, \mathcal{P}) + m\|\mathcal{P}\|_\infty\sqrt{\frac{1}{2m}\log\frac{2}{\delta}}$$

holds with probability at least $1 - \frac{\delta}{2}$.

If the sample $S$ and $\widetilde{S}$ differs only at the $j$-th instance, i.e., $z_i = \widetilde{z}_i$ for all $i \ne j$, then

$$\begin{aligned}
\widehat{\mathscr{R}}_S(\Theta, \mathcal{P}) - \widehat{\mathscr{R}}_{\widetilde{S}}(\Theta, \mathcal{P}) &= \mathbb{E}_{\boldsymbol{\sigma}}\sup_{\theta \in \Theta}\sup_{\boldsymbol{p} \in \mathcal{P}}\left[\sum_{i=0}^{m} \sigma_i p_i \ell(z_i, \theta)\right] - \mathbb{E}_{\boldsymbol{\sigma}}\sup_{\theta \in \Theta}\sup_{\boldsymbol{p} \in \mathcal{P}}\left[\sum_{i=0}^{m} \sigma_i p_i \ell(\widetilde{z}_i, \theta)\right] \\
&\le \mathbb{E}_{\boldsymbol{\sigma}}\sup_{\theta \in \Theta}\sup_{\boldsymbol{p} \in \mathcal{P}}\left[\sigma_j p_j(\ell(z_j, \theta) - \ell(\widetilde{z}_j, \theta))\right] \\
&\le \|\mathcal{P}\|_\infty.
\end{aligned}$$

Similarly, we have $|\widehat{\mathscr{R}}_S(\Theta, \mathcal{P}) - \widehat{\mathscr{R}}_{\widetilde{S}}(\Theta, \mathcal{P})| \le \|\mathcal{P}\|_\infty$. Then the result holds immediately from McDiarmid's inequality. □

**Theorem 4.** *For any $\epsilon > 0$, the empirical Rademacher complexity and the empirical robust Rademacher complexity can be respectively bounded by:*

$$\widehat{\mathscr{R}}_S(\Theta) \leq \sqrt{\frac{2\log\mathcal{N}(\Theta(S), \|\cdot\|, \epsilon)}{m}} + \frac{\epsilon d_*(m)}{m},$$

*and*

$$\widehat{\mathscr{R}}_S(\Theta, \mathcal{P}) \leq (1 + \mathrm{RAD}_2(\mathcal{P}))\sqrt{\frac{2\log\mathcal{N}(\Theta(S, \mathcal{P}), \|\cdot\|, \epsilon)}{m}} + \frac{\epsilon d_*(m)}{m}.$$

*Proof.* First, we introduce a lemma.

**Lemma 7** (Massart 2000). *Let $A$ be a finite set in $\mathbb{R}^m$, with $r = \max_{\boldsymbol{x} \in A} \|x\|_2$, then the following holds:*

$$\mathbb{E}_{\boldsymbol{\sigma}}\left[\sup_{\boldsymbol{x} \in A}\sum_{i=1}^m \frac{\sigma_i x_i}{m}\right] \leq \frac{r\sqrt{2\log|A|}}{m}.$$

Denote $N$ as the $\epsilon$-net of $\Theta(S)$ where $|N| = \mathcal{N}(\Theta(S), \|\cdot\|, \epsilon)$. Notice that $\ell(z_i, \theta) \in [0, 1]$, then $r \leq \sqrt{m}$. Applying Massart's lemma to $N$, we get:

$$\mathbb{E}_{\boldsymbol{\sigma}}\left[\sup_{\boldsymbol{x} \in N}\sum_{i=1}^m \frac{\sigma_i x_i}{m}\right] \leq \sqrt{\frac{2\log\mathcal{N}(\Theta(S), \|\cdot\|, \epsilon)}{m}}.$$

For any $\boldsymbol{x} \in \Theta(S)$, there exists a $\boldsymbol{x}' \in N$, such that $\|\boldsymbol{x} - \boldsymbol{x}'\| \leq \epsilon$. Then

$$\sum_{i=1}^m \frac{\sigma_i x_i}{m} = \sum_{i=1}^m \frac{\sigma_i x_i'}{m} + \sum_{i=1}^m \frac{\sigma_i(x_i - x_i')}{m}$$

$$\leq \sum_{i=1}^m \frac{\sigma_i x_i'}{m} + \frac{\epsilon d_*(m)}{m}.$$

Then

$$\widehat{\mathscr{R}}_S(\Theta) = \mathbb{E}_{\boldsymbol{\sigma}}\left[\sup_{\boldsymbol{x} \in \Theta(S)}\sum_{i=1}^m \frac{\sigma_i x_i}{m}\right]$$

$$\leq \mathbb{E}_{\boldsymbol{\sigma}}\left[\sup_{\boldsymbol{x} \in \Theta(S)}\sum_{i=1}^m \frac{\sigma_i x_i'}{m} + \frac{\epsilon d_*(m)}{m}\right]$$

$$\leq \mathbb{E}_{\boldsymbol{\sigma}}\left[\sup_{\boldsymbol{x}' \in N}\sum_{i=1}^m \frac{\sigma_i x_i'}{m}\right] + \frac{\epsilon d_*(m)}{m}$$

$$\leq \sqrt{\frac{2\log\mathcal{N}(\Theta(S), \|\cdot\|, \epsilon)}{m}} + \frac{\epsilon d_*(m)}{m},$$

which completes the first part of the theorem. The second part is similar, but one should notice that in this case $r = \max\limits_{\boldsymbol{x} \in \Theta(S, \mathcal{P})} \|x\|_2 \leq \sqrt{m}\|\mathcal{P}\|_2 = \sqrt{m}(1 + \mathrm{RAD}_2(\mathcal{P}))$. $\qquad\square$

**Theorem 5.** *Assume the loss function is L-Lipschitz w.r.t. $\theta$ for all $z$. Consider the following hypothesis set:*

$$\Theta_c = \left\{\theta : \mathbb{E}_S\widehat{R}_{S, \mathcal{P}}(\theta) \leq c, \|\theta\|_2 < r\right\}.$$

*For any $\epsilon > 0$, let $\mathcal{N}(\Theta_c, \epsilon)$ denote the covering number of $\Theta_c$. Then for all $0 < \delta < 1$, with probability $\geq 1 - \delta$, the localized empirical Rademacher complexity can be bounded as:*

$$\widehat{\mathscr{R}}_S(\Theta_c) \leq \frac{1}{\sqrt{m}}\left(1 + \frac{4}{\sqrt{m}\mathrm{rad}_\infty(\mathrm{conv}(\mathcal{P}))}\right)\left(c + \epsilon\|\mathcal{P}\|_0\|\mathcal{P}\|_\infty L + \|\mathcal{P}\|_\infty\sqrt{\frac{m}{2}\log\frac{\mathcal{N}(\Theta_c, \epsilon)}{\delta}}\right).$$

*Proof.* Notice that $\ell(z, \theta) \in [0, 1]$ for all $z$ and $\theta$, then if we change a single sample $z_i$ of $S = (z_1, \cdots, z_i, \cdots, z_m)$ to form $S' = (z_1, \cdots, z_i', \cdots, z_m)$, we have

$$\widehat{R}_{S,\mathcal{P}}(\theta) - \widehat{R}_{S',\mathcal{P}}(\theta)$$

$$= \sup_{\boldsymbol{p} \in \mathcal{P}} \widehat{R}_{S,\boldsymbol{p}}(\theta) - \sup_{\boldsymbol{p} \in \mathcal{P}} \widehat{R}_{S',\boldsymbol{p}}(\theta)$$

$$\leq \sup_{\boldsymbol{p} \in \mathcal{P}} (\widehat{R}_{S,\boldsymbol{p}}(\theta) - \widehat{R}_{S',\boldsymbol{p}}(\theta))$$

$$= \sup_{\boldsymbol{p} \in \mathcal{P}} p_i(\ell(z_i, \theta) - \ell(z_i', \theta))$$

$$\leq ||\mathcal{P}||_\infty.$$

By the same argument, we can get $|\widehat{R}_{S,\mathcal{P}}(\theta) - \widehat{R}_{S',\mathcal{P}}(\theta)| \leq ||\mathcal{P}||_\infty$. By McDiarmid's inequality(McDiarmid, 1989), we have:

$$\mathbb{P}\left\{\widehat{R}_{S,\mathcal{P}}(\theta) \geq \mathbb{E}\widehat{R}_{S,\mathcal{P}}(\theta) + t\right\} \leq \exp\left\{\frac{-2t^2}{m||\mathcal{P}||_\infty^2}\right\}.$$

Let $N$ denotes the minimum $\epsilon$-net of $\Theta_c$, where $|N| = \mathcal{N}(\Theta_c, \epsilon)$. Then we have

$$\mathbb{P}\left\{\exists \theta_0 \in N, \widehat{R}_{S,\mathcal{P}}(\theta_0) \geq c + t\right\}$$

$$\leq \mathcal{N}(\Theta_c, \epsilon)\mathbb{P}\left\{\widehat{R}_{S,\mathcal{P}}(\theta) \geq c + t\right\}$$

$$\leq \mathcal{N}(\Theta_c, \epsilon)\mathbb{P}\left\{\widehat{R}_{S,\mathcal{P}}(\theta) \geq \mathbb{E}_S\widehat{R}_{S,\mathcal{P}}(\theta) + t\right\}$$

$$\leq \mathcal{N}(\Theta_c, \epsilon)\exp\left\{\frac{-2t^2}{m||\mathcal{P}||_\infty^2}\right\}.$$

In other words, with probability larger than $1 - \mathcal{N}(\Theta_c, \epsilon)\exp\left\{\frac{-2t^2}{m||\mathcal{P}||_\infty^2}\right\}$, we have

$$\sup_{\theta \in N} \widehat{R}_{S,\mathcal{P}}(\theta) < c + t.$$

By the definition of $\epsilon$-net, for any $\theta \in \Theta_c$, there exists a $\theta' \in N$, such that $||\theta - \theta'|| < \epsilon$. Let $L^S$ denotes an upper bound of the *Lipschitz* constant of $\widehat{R}_S(\theta)$, then we have

$$\sup_{\theta \in \Theta_c} \widehat{R}_{S,\mathcal{P}}(\theta) < c + t + \epsilon L^S, \tag{5}$$

with probability larger than $1 - \mathcal{N}(\Theta_c, \epsilon)\exp\left\{\frac{-2t^2}{m||\mathcal{P}||_\infty^2}\right\}$.

Recall that assuming the loss function $\ell(z, \theta)$ is *L-Lipschitz* with respect to $\theta$ for all $z$, then the *Lipschitz* constant of $\widehat{R}_S(\theta)$ can be bounded by $\|\mathcal{P}\|_0\|\mathcal{P}\|_\infty L$.

For those samples $S = (z_1, \cdots, z_m)$ satisfying the universal bound (5), the empirical Rademacher complexity can be bounded as well. The idea of the proof involves the representation of $\boldsymbol{\sigma}$ as a convex combination of several vectors in $\mathcal{P}$.

$$\widehat{\mathscr{R}}_S(\Theta_c) = \mathbb{E}_{\boldsymbol{\sigma}}\left[\sup_{\theta \in \Theta_c} \sum \frac{\sigma_i \ell(z_i, \theta)}{m}\right]$$

$$\triangleq \mathbb{E}_{\boldsymbol{\sigma}}\left[\sup_{\theta \in \Theta_c} \frac{\boldsymbol{\sigma} \cdot \boldsymbol{\ell}(S, \theta)}{m}\right].$$

Recall that $p_0 = \left(\frac{1}{m}, \cdots, \frac{1}{m}\right)^\top$ and by our assumption of the symmetry of $\mathcal{P}$, we have $\boldsymbol{p}_0 \in \text{conv}(\mathcal{P})$. We hope that

$$\frac{\boldsymbol{\sigma}}{m} = a\boldsymbol{p}_0 + b(\boldsymbol{p}_1 - \boldsymbol{p}_0),$$

where $p_1 \in \mathrm{conv}(\mathcal{P})$. Summing up all the coordinates of both sides of the equation, we have:

$$\frac{\sum \sigma_i}{m} = a\mathbf{1} \cdot p_0 + b\left(\mathbf{1} \cdot p_1 - \mathbf{1} \cdot p_0\right) = a.$$

Define $\bar{\sigma} = \dfrac{\sum \sigma_i}{m}$, Then

$$b(p_1 - p_0) = \left(\frac{\sigma_1}{m} - \frac{\bar{\sigma}}{m}, \cdots, \frac{\sigma_m}{m} - \frac{\bar{\sigma}}{m}\right)^\top. \tag{6}$$

To derive the optimal bound, take the $L_\infty$ norm of both sides of (6), we get

$$|b| \|p_1 - p_0\|_\infty = \max\left\{|\frac{\sigma_1}{m} - \frac{\bar{\sigma}}{m}|, \cdots, |\frac{\sigma_m}{m} - \frac{\bar{\sigma}}{m}|\right\} \le \frac{2}{m}.$$

Notice that $\|p_1 - p_0\|_\infty \ge \mathrm{rad}_\infty(\mathrm{conv}(\mathcal{P}))$, so $|b| \le \dfrac{2}{m\,\mathrm{rad}_\infty(\mathrm{conv}(\mathcal{P}))}$.

Summarizing our result, we have:

$$\frac{\sigma}{m} \cdot \ell(S, \theta) = [\bar{\sigma}p_0 + b(p_1 - p_0)] \cdot \ell(S, \theta)$$
$$\le |\bar{\sigma} - b| |p_0 \cdot \ell(S, \theta)| + |b| |p_1 \cdot \ell(S, \theta)|$$
$$\le (|\bar{\sigma}| + 2|b|) \widehat{R}_{S,\mathcal{P}}(\Theta_c).$$

Then

$$\widehat{\mathscr{R}}_S(\Theta_c) \le \mathbb{E}_{\sigma} \left(|\bar{\sigma}| + 2|b|\right) \widehat{R}_{S,\mathcal{P}}(\Theta_c)$$
$$\le \left(\sqrt{\mathbb{E}_{\sigma} |\bar{\sigma}|^2} + \frac{4}{m\,\mathrm{rad}_\infty(\mathrm{conv}(\mathcal{P}))}\right) \widehat{R}_{S,\mathcal{P}}(\Theta_c)$$
$$= \frac{1}{\sqrt{m}} \left(1 + \frac{4}{\sqrt{m}\,\mathrm{rad}_\infty(\mathrm{conv}(\mathcal{P}))}\right) \widehat{R}_{S,\mathcal{P}}(\Theta_c).$$

$\square$

## C  THE DETAILS OF WSGD

We denote the vector $v(q, r) = (v_1, v_2, \ldots, v_m)$:

$$v_i = \frac{1}{\widetilde{W}} \left(r \sum_{q|B_t| > j \ge 0} C_{m-i}^j C_{i-1}^{|B_t|-j-1} + \sum_{|B_t| > j \ge q|B_t|} C_{m-i}^j C_{i-1}^{|B_t|-j-1}\right)$$

where $\widetilde{W}$ is the normalizer $s.t.$ $\sum_{i=1}^m v_i = 1$. Then the stochastic gradient calculated in $\mathrm{WSGD}(q, r)$ is an unbiased estimation of the gradient of $\widehat{R}_{S,\mathcal{P}(v(q,r))}(\theta)$ at differentiable point. Recall that $\sigma(v)$ is the set of all the vectors whose entries are permutations of $v$'s and $\mathcal{P}(v)$ denote the convex hull of $\sigma(v)$.

