# OpenReview forum: "Distributionally Robust Optimization Leads to Better Generalization: on SGD and Beyond"
_ICLR.cc/2019/Conference_

### Official Review · AnonReviewer1 · 2018-11-05
**Interesting connection between SGD and DRO, but writing and experiments need more clarity**

**Rating:** 5
**Confidence:** 4

**Review:**


This paper consider the connections between SGD and distributionally robust optimization. There has long been observed a connection between robust optimization and generalization. Recently, this has been explored through the lens of distributionally robust optimization. e.g., in the papers of Namkoong and Duchi, but also many others, e.g., Farnia and Tse, etc. Primarily, this paper appears to build off the work of Namkoong.

The key connection this paper tries to make is between SGD and DRO, since SGD in sampling a minibatch, can be considered a small perturbation to the distribution. Therefore the authors use this intuition to propose a weighted version of SGD (WSGD) whereby high variance weights are assigned to mini batch, thus making the training accomplish a higher level of distributional robustness.

This idea is tested on a few data sets including CIFAR-10 and -100. The results compare WSGD with SGD, and they show that the WSGD-trained models have a lower robust loss, and also have a higher (testing) accuracy.

This is an interesting paper. There has been much discussion of the role of batch size, and considering it from a different perspective seems to be of interest. But the connection of the empirical results to the theoretical results seems tenuous. It’s not clear how predictions of the theory match up. This would be useful to understand better. More generally, a simpler presentation of the key results would be useful, so as to allow the reader to better appreciate what are the main claims and if they are as substantial as claimed. Overall the writing needs significant polishing, though this is only at a local level, i.e, it doesn’t obscure the flow of the paper.

---

> ### Author Response · Authors · 2018-11-26
> **Response to Reviewer**
>
> Thank you very much for your review and comments! We apologize for the writing and we will polish it. Follow-up work is proceeding and hopefully we will provide more connection of the theoretical results to the empirical result.

---

### Official Review · AnonReviewer3 · 2018-11-12
**Good motivation but shaky theory, disconnected algorithm, and weak experiments.**

**Rating:** 4
**Confidence:** 4

**Review:**

This paper motivates performing a “robustified” version of SGD. It attempts to formalize this notion, proves some variants of generalization bounds, and proposes an algorithm that claims to implement such a modified SGD.

The clarity of the paper can be improved. There are several notational and language ambiguities throughout. Most importantly, a couple of paragraphs that are meant to convey key intuitions about the results are very badly written (the one following Theorem 1, the one preceding Theorem 2, the last one in Section 5.1, and the one preceding Theorem 5, more on these later).

Apart from these clarity issues, the significance of the results is weak. This is because although the technical statements seem correct, the leap from them to measurable outcomes (such as actual generalization bounds) are missing. Part of this is due to a lack of a good notion of “true” robust risk. Moreover the algorithmic contribution does not connect well with the suggested theory and the experimental results are modest at best. Here is a more detailed breakdown of my objections.

-	The notion of distributional robust loss is sound, i.e. R(\theta, K). Its empirical variant is also good, \hat{R}(\theta, K). But the notion of robust loss defined in the paper, \hat R_{S,P}(\theta) with the weights on the samples, breaks from this notion. The reason is that in the former case the weights depend on the value of the sample (z) whereas in the latter they depend on the index (i). It is not evident how to map one to the other.

-	This makes the question of what is the “true” robust risk unclear. It is tempting to simply say it is its expectation with respect to a generic sample. This is the view taken in Theorem 3, which offers a kind of generalization bound. But if one looks carefully at this, the location of the expectation and supremum should be swapped. Here is an alternative view: if want to think of the infinite sample limit, then we need to have a sequence of robustness classes P_m, that vary with m (say those that put weight only on a q-fraction of the samples, just like in the suggested WSGD). The “true” robust risk would be the limit of the sup of the empirical risk, this keeps the sup and expectation in the right order. Under the right conditions, this limit would indeed exist. And it is difficult to know, for a given m, how *far* the generic-sample expectation of Theorem 3 is from it. Without this knowledge, it is difficult to interpret Theorem 3 as a generalization bound.

-	Theorem 1 itself is a standard result. The discussion after Theorem 1 is the kind of argument that also explains Fisher information, and can be presented more clearly. I’m not sure whether Theorem 2 exactly proves what the paragraph before it is trying to explain. The fact that SGD converges into the ball seems contrived, since the quantities that we are trying to bound have nothing to do with the optimization method. If the optimum is within the ball (+/- something) then the same result should hold with the step size replaced with the (+/- something). So how does this explain escaping stationary points?

-	If we accept Theorem 3 as a generalization bound, alongside with the Rademacher bounds of Theorem 4, I don’t think the paper treats the balance between the various terms adequately enough. In particular we see that the |P|_\infty term in Theorem 3 has to balance out the (robust) Rademacher bound, and need it to be of the order of (1+RAD_2(P)\sqrt{\log N}/m). For P that puts weight k over 1/k points, |P|_\infty = 1/k. RAD_2(P) is bounded by 1/\sqrt{k}, so it’s negligible next to 1. But the covering number N can grow exponentially with k (when it’s not too large, and for small \epsilon, just by counting arguments). So this seems to say that for a good tradeoff in the bounds will lead to k having to be a growing fraction of m. This intuition, if true, is not presented. Not only that, but it also goes against the suggested approach of choosing some constant fraction of m.

-	Theorem 5 gives a local Rademacher complexity. But again there is a conceptual step missing from this to strong generalization bounds, partly because we are not exactly minimizing the empirical risk within the considered class. Also, the discussion that follows bounding the rad_\infty with |P|_\infty is deficient, because it misses again the fact there are two salient terms that need to balance out.

-	Algorithm 1 (WSGD) needs to specify (q,r) and which G (G_1 or G_2) as inputs too.

-	Most importantly, WSGD does not seem to be minimizing the robust risk at all. First, I’m not really sure what the G_1 variant does. If we were to follow the intuition of Theorem 1, we should be looking at the gradients, not the loss values. As for G_2, by sampling we are in fact replacing the sup over p with an average over P. This can have a significantly different behavior, and we could possibly interpret it as a slightly reduced effective batch size, especially in the case of G_2. In fact, in the experiments, when r is set to 0, this is exactly what is happening! At any rate, it is not clear at all how any of the earlier sections connect with sections 6 or 7.

-	In the experimental section it is not clarified which of the latter two is used (I assume G_2, the randomized one, given the discussion in the end of Section 6.) When the authors write “accuracy improvement”, they should more clearly say “relative decrease in misclassification error”. That’s the only thing that makes sense with the numbers, and if it does in fact  the authors mistakenly say that the 5-15% improvement is for CIFAR 100 and the 5% is for CIFAR 10, it’s the other way around! And the exception (least) improvement seems to be ResNet-34 on CIFAR-100 (not VGG-16, as they claim, unless the table is wrong.) All in all, these are all pretty weak results, albeit consistent. A better benchmark would have been to compare against various batch sizes, and somehow show that the results do *not* follow from batch size effects.

---

> ### Author Response · Authors · 2018-11-24
> **Response to Reviewer (Part 1/3)**
>
> We thank the reviewer for their comments. However, there are some misunderstandings that need to be corrected.
>
> -The notion of distributional robust loss is sound, i.e. R(\theta, K). Its empirical variant is also good, \hat{R}(\theta, K). But the notion of robust loss defined in the paper, \hat R_{S,P}(\theta) with the weights on the samples, breaks from this notion. The reason is that in the former case the weights depend on the value of the sample (z) whereas in the latter they depend on the index (i). It is not evident how to map one to the other.
>
> For fixed sample S, we denote P(S, K)={(P_\lambda(z_1)/P_\lambda_0(z_1),...,P_\lambda(z_m)/P_\lambda_0(z_m)): \lambda \in K}, which maps K into a weight set. Since we only have access to the training data, and the underlying distribution is unknown, it's impractical to use R(\theta,K) or \hat{R}(\theta,K) to develop algorithms. So we seek some substitutes, and \hat R_{S,P}(\theta) is a good one. It seems to assign weights according to the indices, but it's not the case. We have assumed P is symmetrical, and when the supremum is taken over all possible weights, it doesn't matter how you arrange the instances (z_i).
>
>
> It's true that we assume the weights independent of the value of the sample, so that it's convenient to analyze the generalization properties afterwards.
>
>
>
> -I'm not sure whether Theorem 2 exactly proves what the paragraph before it is trying to explain. The fact that SGD converges into the ball seems contrived, since the quantities that we are trying to bound have nothing to do with the optimization method. If the optimum is within the ball (+/- something) then the same result should hold with the step size replaced with the (+/- something). So how does this explain escaping stationary points?
>
> What we want to argue in Theorem 2 is that when SGD converges with large step size, the robust loss can be bounded. This has nothing to do with escaping from saddle point.
>
>
>
> -For P that puts weight k over 1/k points, |P|_\infty = 1/k. RAD_2(P) is bounded by 1/\sqrt{k}, so it's negligible next to 1. But the covering number N can grow exponentially with k (when it's not too large, and for small \epsilon, just by counting arguments). So this seems to say that for a good tradeoff in the bounds will lead to k having to be a growing fraction of m. This intuition, if true, is not presented. Not only that, but it also goes against the suggested approach of choosing some constant fraction of m.
>
> It's helpful to notice that the set conv(P_k) decreases monotonically when k increases. We don't agree that the covering number N(\Theta(S,P)) grows exponentially with k, instead the converse might be true. But in my point, analyzing the convergence rate of the upper bound is meaningless. We are in favour of DRO because it can achieve better results when data are limited and the model capacity is excessive.

---

> ### Author Response · Authors · 2018-11-24
> **Response to Reviewer (Part 2/3)**
>
> -This makes the question of what is the "true" robust risk unclear. It is tempting to simply say it is its expectation with respect to a generic sample. This is the view taken in Theorem 3, which offers a kind of generalization bound. But if one looks carefully at this, the location of the expectation and supremum should be swapped. Here is an alternative view: if want to think of the infinite sample limit, then we need to have a sequence of robustness classes P_m, that vary with m (say those that put weight only on a q-fraction of the samples, just like in the suggested WSGD). The "true" robust risk would be the limit of the sup of the empirical risk, this keeps the sup and expectation in the right order. Under the right conditions, this limit would indeed exist. And it is difficult to know, for a given m, how *far* the generic-sample expectation of Theorem 3 is from it. Without this knowledge, it is difficult to interpret Theorem 3 as a generalization bound.
> -Theorem 5 gives a local Rademacher complexity. But again there is a conceptual step missing from this to strong generalization bounds, partly because we are not exactly minimizing the empirical risk within the considered class.
>
> Your confusion about Theorem 5 just provides the solution of your previous question. Theorem 3 doesn't aim to provide a bound on generalization error R(\theta) or distributional robust loss R(\theta, K) at all. It aims to provide guarantees that minimizing empirical robust loss leads to a small expected robust loss w.r.t. a generic sample. This is fundamental to the analysis on the localized Rademacher complexity in Section 5.2, since the class \Theta_c we have considered requires the expected robust loss is lower than c. The alternative view you provided is quite right, but not the topic we studied in this paper. We mainly focus on how the size of P influences the generalization when m is fixed. The behavior when P_m varies with m is meaningful, but it requires us to appoint how P_m varies with m. However, we did not want to specify the shape of P in this paper.
>
> Various generalization bounds on R(\theta) in terms of the empirical robust loss have been developed in previous works.
>
>
>
> -Also, the discussion that follows bounding the rad_\infty with |P|_\infty is deficient, because it misses again the fact there are two salient terms that need to balance out.
>
> Note that \epsilon in Theorem 5 can be chosen arbitrarily. If we choose \epsilon=O(1/m^k), it will just add a term of O(ln m) to the last term.  We did not find the two salient terms you have mentioned. Can you specify them?

---

> ### Author Response · Authors · 2018-11-24
> **Response to Reviewer (Part 3/3)**
>
> -Algorithm 1 (WSGD) needs to specify (q,r) and which G (G_1 or G_2) as inputs too.
> -Most importantly, WSGD does not seem to be minimizing the robust risk at all. First, I'm not really sure what the G_1 variant does. If we were to follow the intuition of Theorem 1, we should be looking at the gradients, not the loss values. As for G_2, by sampling we are in fact replacing the sup over p with an average over P. This can have a significantly different behavior, and we could possibly interpret it as a slightly reduced effective batch size, especially in the case of G_2. In fact, in the experiments, when r is set to 0, this is exactly what is happening! At any rate, it is not clear at all how any of the earlier sections connect with sections 6 or 7.
> -In the experimental section it is not clarified which of the latter two is used (I assume G_2, the randomized one, given the discussion in the end of Section 6.)
>
> We admit that there is some abuse of notation. We have pointed out that in the algorithm section, G_1 is used. We proposed a framework of algorithm called WSGD. When we talk about WSGD(q,r), it refers in particular to WSGD equipped with G_1(q,r). All the theoretical analysis and experiments below are based on G_1! WSGD(q,r) calculated the stochastic gradient of a robust loss (see Appendix C), and obviously it minimizes the robust loss. So the effect of WSGD(q,r) is far beyond just reducing the batch size. We gave some discussion on G_2 in the end of Section 6 because it may work well in some tasks, which we did no™t involve in this paper. We implore you to read Sections 6 and 7 again, because there appears some misunderstanding in the previous reading.
>
>
>
> -When the authors write "accuracy improvement", they should more clearly say "relative decrease in misclassification error". That's the only thing that makes sense with the numbers, and if it does in fact  the authors mistakenly say that the 5-15\% improvement is for CIFAR 100 and the 5\% is for CIFAR 10, it's the other way around!
>
> We agree.  We have corrected the mistake in the new version.
>
>
>
> -A better benchmark would have been to compare against various batch sizes, and somehow show that the results do *not* follow from batch size effects.
>
> Again, we wish to emphasize that WSGD(q,r) is WSGD equipped with G_1, so the role of WSGD(q,r) is not just reducing the batch size. Furthermore, empirically speaking, the batch size almost disappears when the batch size is below 128, i.e., the model trained with batch size=32, 64, 128 always have the same performance). So we compared SGD and WSGD(q,r) with batch size 128 for variable control.

---

### Official Review · AnonReviewer2 · 2018-11-20
**The paper needs major revisions, theorems are disjoint with few explanations.**

**Rating:** 3
**Confidence:** 3

**Review:**

The paper aims to connect "distributionally robust optimization" (DRO) with stochastic gradient descent. The paper purports to explain how SGD escapes from bad local optima and purports to use (local) Rademacher averages (actually, a  generalization defined for the robust loss) to explain the generalization performance of SGD.

In fact, the paper proves a number of disjointed theorems and does very little to explain the implications of these theorems, if there are any. The theorem that purports to explain why SGD escapes bad local minima does not do this at all. Instead, it gives a very loose bound on the "robust loss" under some assumptions that actually rule out ReLU networks.

The Rademacher results for robust loss looked promising, but there is zero analysis suggesting why these explain anything. Instead, there is vague conjecture. The same is true for the local Rademacher statements. It is not enough to prove a theorem. One must argue that it bears some relationship to empirical performance and this is COMPLETELY missing.

Other criticisms:

1. One of the first issues to arise is that the definition of "generalization error" is not the one typically used in learning theory. Here generalization error is used for what is more generally called the risk.  Generalization error often refers to the difference R(theta) - ^R(theta) between the risk and the empirical risk (i.e., the risk evaluated against the empirical distribution). (Generally this quantity is positive, although sometimes its absolutely values is bounded instead.)

Another issue with the framing is that one is typically not interested in small risk in absolute terms, but instead small risk relative to the best risk available in some class (generally the same one that is being used as a source of classifiers). Thus one seeks small excess risk. I'm sure the authors are aware of these distinctions, but the slightly different nomenclature/terminology may sow some confusion.

2. The unbiased estimate suggested on page 2 is not strictly speaking an estimator because it depends on \lambda_0, which is not measurable with respect to the data. The definition of K and how it relates to the estimate \hat \lambda is vague. Then the robust loss is introduced where the unknown quantity is replaced by a pre-specified collection of weights. If these are pre-specified (and not data-dependent), then it is really not clear how these could be a surrogate for the distribution-dependent weights appearing in the empirical distributionally robust loss.

Perhaps this is all explained clearly in the literature introducing DRO, but this introduction leaves a lot to be desired.

3. "This interpretation shows a profound connection between SGD and DRO." This connection does not seem profound to a reader at this stage of the paper.

4. Theorem 2 seems to be far too coarse to explain anything. The step size is very small and so 1/eta^2 is massive. This will never be controlled by 1/mu, and so this term alone means that there is affectively no control on the robust loss in terms of the local minimum value of the empirical risk.

5. There seems to be no argument that robustness leads to any improvement over nonrobust... at least I don't see why it must be true looking at the bounds. At best, an upper bound would be shown to be tighter than another upper bound, which is meaningless.


Corrections and typographical errors:

1. There are grammatical errors throughout the document. It needs to be given to a copy editor who is an expert in technical documents in English.

2. "The overwhelming capacity ... of data..." does not make sense. The excessive complexity of the sentence has led to grammatical errors.

3. The first reference to DRO deserves citation.

4. It seems strange to assume that the data distribution P is a member of the parametric model M. This goes against most of learning theory, which makes no assumption as to the data distribution, other than the examples being i.i.d.

5. You cite Keskar (2016) and Dinh (2017) around sharp minima. You seem to have missed Dziugaite and Roy (2017, UAI) and Neyshabur et al (NIPS 2017), both of which formalize flatness and give actual generalization bounds that side step the issue raised by Dinh.

6. "not too hard compared" ... hard?

7. Remove "Then" from "Then the empirical robust Rademacher...". Also removed "defined as" after "is".

8. "Denote ... as an" should be "Let ... denote the..." or "Denote by ... the upper ..."

9. " the generalization of robust loss is not too difficult" ... difficult?

10. "some sort of “solid,” " solid?

11. "Conceivably, when m and c are fixed, increasing the size of P reduces the set Θc". Conceivably? So it's not necessarily true? I don't understand the role of conceivably true statements in a paper.

[This review was requested late in the process due to another reviewer dropping out of the process.]

[UPDATE] Authors' response to my questions did not change my opinion about the overall quality of the paper. Both theory and writing need a major revision.

---

> ### Author Response · Authors · 2018-11-24
> **Response to Reviewer**
>
> Thank you for the review. Here are our response to your specific comments:
>
> -One of the first issues to arise is that the definition of ``generalization error‘’ is not the one typically used in learning theory. Here the generalization error is used for what is more generally called the risk.  Generalization error often refers to as the difference R(theta) - ^R(theta) between the risk and the empirical risk (i.e., the risk evaluated against the empirical distribution). (Generally this quantity is positive, although sometimes the absolute value is bounded instead.)
>
> There are many machine learning literatures which refer to the term R(\theta) as generalization error, such as Foundations of Machine Learning (Mohri et al.).
>
> It's clear that when the class of models is fixed, small risk is equivalent to small excess risk.
>
>
>
> -The unbiased estimate suggested on page 2 is not strictly speaking an estimator because it depends on \lambda_0, which is not measurable with respect to the data. The definition of K and how it relates to the estimate \hat \lambda is vague. Then the robust loss is introduced where the unknown quantity is replaced by a pre-specified collection of weights. If these are pre-specified (and not data-dependent), then it is really not clear how these could be a surrogate for the distribution-dependent weights appearing in the empirical distributionally robust loss.
>
> We admit that, strictly speaking, it's not an estimator, but it doesn't obstruct the idea. Actually the weight set P is data dependent. When the sample is fixed, we consider the robust loss with a pre-specified P. We replace the unknown quantity by a pre-specified collection of weights simply because K is unknown and we seek some surrogate (the set P) in hope to cover those weights in \hat{R}(\theta,K). When the dataset changes, we need to choose another weight set. The weight set P can be viewed as a tuning parameter.
>
>
>
> -Perhaps this is all explained clearly in the literature introducing DRO, but this introduction leaves a lot to be desired.
>
> We have carefully investigate other works introducing DRO, and many define the weight set P to be all the weights p that is "close" to uniform distribution (p_0) using f-divergence or \phi-divergence. Our approach is a natural generalization of theirs.
>
>
>
> -Theorem 2 seems to be far too coarse to explain anything. The step size is very small and so 1/eta^2 is massive. This will never be controlled by 1/mu, and so this term alone means that there is affectively no control on the robust loss in terms of the local minimum value of the empirical risk.
>
> The term 1/eta^2 can be controlled by B^2. What we want to argue in Theorem 2 is that when SGD converges with large step size, the robust loss can be bounded. So the step size is not very small actually, and the term (B/eta)^2 measures how well SGD converges.
>
>
>
> - There seems to be no argument that robustness leads to any improvement over nonrobust... at least I don't see why it must be true looking at the bounds. At best, an upper bound would be shown to be tighter than another upper bound, which is meaningless.
>
> The bound in Theorem 5 shows how the size of P influences the local Rademacher complexity. From what we have seen so far, we are the first to build such a bound and whereas no other upper bound at all.
>
>
> ---------------
> -It seems strange to assume that the data distribution P is a member of the parametric model M. This goes against most of learning theory, which makes no assumption as to the data distribution, other than the examples being i.i.d.
>
> This part just shows our motivation, and for simplicity and clarity we introduce a parametric family M. An equivalent argument without referring to M is also possible.
>
>
>
> -"Conceivably, when m and c are fixed, increasing the size of P reduces the set Θc". Conceivably? So it's not necessarily true? I don't understand the role of conceivably true statements in a paper.
>
> The statement is true in the following sense: if P1 is a subset of P2, then \Theta_c(P2) is a subset of \Theta_c(P1). The result  immediately follows from the definitions, so we omit the proof.

---

### Public Comment · ~Weihua_Hu1 · 2018-09-30
**DRO applied to classification**

Our ICML 2018 paper "Does Distributionally Robust Supervised Learning Give Robust Classifier?" analyzed DRO applied to classification and showed several negative results about it. The key to show our results is that in classification, the 0-1 loss is used for testing, which is different from a surrogate loss used for training.
Link: http://proceedings.mlr.press/v80/hu18a/hu18a.pdf

It would be great to have discussion on our paper as your empirical evaluation is on classification tasks.

---

> ### Author Response · Authors · 2018-10-01
> **Thank you for the feedback**
>
> Thank you for the feedback. We have carefully read your work before, but we think there is no conflict between your work and ours. In our understanding, your work compared ERM and DRO with 0-1 loss and with surrogate loss. However, in our paper we don’t deal with ERM loss at all.
>
> It is true that in classification, the 0-1 loss is used for testing, which is different from a surrogate loss used for training. However, the problem you pointed out is also a problem for ERM. If we write a article “Does Empirical Risk Minimization Give Effective Classifier?”, how will you response us? In classification, we normally think that optimizing an object function with surrogate loss such as cross-entropy will give a solution that performs well when tested with 0-1 loss. Last but not least, our algorithm really performs better than normal SGD on classification tasks in our experiment.

---

> > ### Public Comment · ~Weihua_Hu1 · 2018-10-01
> > **Reply**
> >
> > For sure, I agree that ERM with a surrogate loss empirically gives a good classifier when tested with the 0-1 loss, which is also supported by the theory of classification calibration loss.  What our paper shows is that DRO applied to classification gives (almost) the same classifier as the ERM (in terms of the 0-1 loss). Empirically, DRO can even be worse than the ERM because it puts large weights on large losses and thus can be extremely sensitive to outliers.
> >
> > I trust your empirical results as well as your theoretical results, but what I want to emphasize here is that there is a huge gap between "small surrogate loss" and "small 0-1 loss (what we really care about)", and we need to be especially careful about this gap when dealing with DRO applied to classification!

---

> > > ### Author Response · Authors · 2018-10-02
> > > **Reply**
> > >
> > > I get your point. Actually we have different views on the data with large losses. The steeper loss function you constructed in your paper can be viewed as a loss function enlarging the gradient (and thus put large weights) of data with large loss. You think that data with large loss may be outliers and can do harm to our model. But I think that large losses reflect a drawback of the model and thus we should pay more attention to those data. In fact,  both of the views can be true in practice, but we can't distinguish without extra domain knowledge. The main idea of our paper is that if the sample is of high quality (few outliers), we should pay more attention to individual instance. By the way, the algorithm in our paper is not based on f-divergence, and the steeper loss may be not applicable to our algorithm.

---

> > > > ### Public Comment · ~Weihua_Hu1 · 2018-10-02
> > > > **Reply**
> > > >
> > > > > By the way, the algorithm in our paper is not based on f-divergence, and the steeper loss may be not applicable to our algorithm.
> > > > Our steepness result holds as long as the larger weights are assigned to larger losses. However, I am not sure what will happen if the mini-batch losses are reweighted, which is the focus of your paper.
> > > >
> > > > >  The main idea of our paper is that if the sample is of high quality (no outliers), we should pay more attention to individual instance.
> > > > I see your point, but I have to say that that is a very strict assumption considering the real-world applications.
> > > >
> > > > >  But I think that large losses reflect a drawback of the model and thus we should pay more attention to those data.
> > > > As DNNs often overfit the training data (achieve 0 training error) eventually, why do we need to pay attention to large loss data during training?
> > > > Even when DNNs do not have enough capacity to overfit the training data, paying more attention to large surrogate loss will inevitably increase the surrogate loss of other data, which may increase the overall misclassification rate (measured by 0-1 loss). What do you think about this point?

---

> > > > > ### Author Response · Authors · 2018-10-02
> > > > > **Reply**
> > > > >
> > > > > In fact, we just take advantage of of the capacity of DNNs. Due to the strong capacity of DNNs, we can pay more attention to large surrogate loss without increasing the surrogate loss of other data too much.
> > > > >
> > > > > We have clarified the necessity to do DRO in our introduction section. I agree that there exists some outliers in the dataset. However, if you want to talk about outliers, you have to introduce extra domain knowledge. Suppose you have trained a model using the dataset, you can't say that the data with large loss (calculated by the model you trained) are outliers. They can be high-quality data which you didn't fit well and the real outliers can have small loss. It is entirely possible that to fit the large loss data is doing regularization on the model and alleviate overfitting.
> > > > >
> > > > > The last thing I want to say is the number of outliers. Strictly speaking, it depends on your measure. In our experiment, we found that about 0.01%-0.1% of the data in CIFAR are outliers. However, what we actually do by DRO is to pay more attention to the data with top 10% or 50% (roughly speaking) loss, and the effect of outliers is negligible. Actually what we do have achieved a good performance.

---

> > > > > > ### Public Comment · ~Weihua_Hu1 · 2018-10-02
> > > > > > **Thanks!**
> > > > > >
> > > > > > I see your points. Thanks for your discussion!
> > > > > >
> > > > > > I think it useful to have training curves in your paper, comparing SGD and WSGD. Intuitively, WSGD may speed up the convergence, but SGD should also achieve 0 training loss eventually. If this is the case, it is mysterious to me why WSGD generalizes better than SGD. Maybe there is inductive bias in WSGD?
> > > > > >
> > > > > > Also, CIFAR is just a benchmark dataset; that's why it is clean.

---

> > > > > > > ### Author Response · Authors · 2018-10-02
> > > > > > > **Reply**
> > > > > > >
> > > > > > > We sincerely thank you for your discussion. We haven't provided the training curve in our paper because of the paper length. Empirically speaking, WSGD doesn't markedly speed up the convergence, but it achieves a lower robust loss eventually (we use cross-entropy loss).
> > > > > > >
> > > > > > > The point we want to emphasize in this paper is that an optimization algorithm in deep learning is not only an optimizer but also a minima selector. We propose WSGD because it has better performance in minima selecting.
> > > > > > >
> > > > > > > In the end, we wish to express our thanks for your patient discussion again!

---

> > > > > > > > ### Public Comment · ~Weihua_Hu1 · 2018-10-02
> > > > > > > > **Thanks!**
> > > > > > > >
> > > > > > > > Thanks for your discussion!
> > > > > > > > The aspect of minima selector is something our paper did not consider; we basically considered the convex scenario and provided analyses for the global solution of DRO.
> > > > > > > >
> > > > > > > > I think it would be extremely helpful to have some discussion on our paper and make the distinction clear. Thanks!

---

### Meta-Review · Area_Chair1 · 2018-12-16
**Needs rewrite**

**Confidence:** 5
**Recommendation:** Reject

**Metareview:**

This paper received high quality reviews, which highlighted numerous issues with the paper.  A common criticism was that the results in the paper seemed disconnected.  Numerous technical concerns were raised. Reading the responses, it seems that some of these issues are nonissues, but it seems also that the writing was not sufficiently up to the standard required of this type of technical work. I suggest the authors produce a rewrite and resubmit to the next ML conference, taking the criticisms they've received here very seriously.